# The right hippocampus leads the bilateral integration of gamma-parsed lateralized information

Nuria Benito[1†‡§], Gonzalo Martín-Vázquez[1†], Julia Makarova[1,2†], Valeri A Makarov[2,3*], Oscar Herreras[1*]

[1]Department of Translational Neuroscience, Cajal Institute – CSIC, Madrid, Spain; [2]N.I. Lobachevsky State University of Nizhny Novgorod, Nizhny Novgorod, Russia; [3]Department of Applied Mathematics, Faculty of Mathematics, Universidad Complutense de Madrid, Madrid, Spain

*For correspondence: vmakarov@mat.ucm.es (VAM); herreras@cajal.csic.es (OH)

[†]These authors contributed equally to this work

Present address: [‡]Institute of Physiology, University Medical Center of the Johannes Gutenberg University, Mainz, Germany; [§]Institute of Patophysiology, University Medical Center of the Johannes Gutenberg University, Mainz, Germany

Competing interests: The authors declare that no competing interests exist.

**Abstract** It is unclear whether the two hippocampal lobes convey similar or different activities and how they cooperate. Spatial discrimination of electric fields in anesthetized rats allowed us to compare the pathway-specific field potentials corresponding to the gamma-paced CA3 output (CA1 Schaffer potentials) and CA3 somatic inhibition within and between sides. Bilateral excitatory Schaffer gamma waves are generally larger and lead from the right hemisphere with only moderate covariation of amplitude, and drive CA1 pyramidal units more strongly than unilateral waves. CA3 waves lock to the ipsilateral Schaffer potentials, although bilateral coherence was weak. Notably, Schaffer activity may run laterally, as seen after the disruption of the connecting pathways. Thus, asymmetric operations promote the entrainment of CA3-autonomous gamma oscillators bilaterally, synchronizing lateralized gamma strings to converge optimally on CA1 targets. The findings support the view that interhippocampal connections integrate different aspects of information that flow through the left and right lobes.

## Introduction

Lateralization of certain neural functions in vertebrates is thought to bear evolutionary advantages (*Halpern et al., 2005*). Earlier studies have mainly focused on finding anatomical correlates to behavioral asymmetries, but there has been little insight gained into the functional mechanisms underlying the differential routing and integration of activity in bilateral networks. For example, fMRI studies have shown bilateral or lateral activation of the same structures when subject performs different tasks (*Smith and Goodale, 2015*; *Lee et al., 2016*).

The bilateral function in the hippocampus is largely unknown and it is unclear to what extent the two lobes convey similar or complementary information, and whether they do indeed work together. However, the existence of important bilateral connections between the same and different hippocampal subregions does suggest some degree of integration and cooperation. In rodents, hippocampal lateralization is observed during certain memory tasks (*Klur et al., 2009*; *Shipton et al., 2014*), in the expression of synaptic plasticity (*Kohl et al., 2011*), or following environmental enrichment (*Shinohara et al., 2013*). In the human, such lateralization was reported during sequence disambiguation (*Kumaran and Maguire, 2006*) and in cognitive navigation (*Iaria et al., 2003*; *Iglói et al., 2015*). Available data is however limited and it provides no mechanistic insights. Here we have used pathway-specific local field potentials (LFPs) (*Herreras et al., 2015*) in anesthetized rats to study the spontaneous transmission in the bilateral CA3→CA1 Schaffer segment during

**eLife digest** In humans and other backboned animals, the brain is divided into the left and right hemispheres, which are connected by several large bundles of nerve fibers. Thanks to these fiber tracts, sensory information from each side of the body can reach both sides of the brain. However, although many areas of the brain work with a counterpart on the opposite hemisphere to process this sensory information, they do not necessarily perform the same tasks, or perform them at the same time as their partner.

The hippocampus is a brain region that helps to support navigation, to detect novelty, and to produce memories. In fact, our brains contain two hippocampi – one in each hemisphere. Previous studies of the hippocampus have tended to record from only one side of the brain. Benito, Martín-Vázquez, Makarova et al. now compare the activity of the left and right hippocampi, and consider how the two structures might work together.

Recordings of the electrical activity of the hippocampi of anesthetized rats show that different groups of neurons fire in rhythmic sequence, forming waves called gamma waves. Successive waves have different amplitudes, and can be thought to form 'strings'. The recordings made by Benito et al. show that the two hippocampi produce parallel strings of waves, although the waves that originate in the right hemisphere are generally larger than those that originate in the left. Right-hemisphere waves also tend to begin slightly earlier than their left-hemisphere counterparts.

Further experiments revealed that disrupting the fiber tracts between the hemispheres uncouples the waves that no longer occur at the same time, and the strings of waves may remain constrained to one side of the brain. In healthy animals, however, the right-hand dominance acts as a master-slave device, and makes the waves from the two hemispheres pair up and merge in the neurons that receive them both. Thus the information running in both hippocampi can be integrated or compared before sending to the cortex for task execution or storage.

Overall, the findings reported by Benito et al. suggest that different types of information flow through the left and right hemispheres, and that the brain integrates these two streams using asymmetric connections. The next challenge is to identify how the information in the two streams differs: whether each stream reflects different sensory stimuli, different features of a scene, or the difference between recalled and perceived information.

irregular non-theta EEG states, a pattern that in active animals is associated to sensory input during immobility and consummatory behaviors.

The hippocampal CA3 region is an important hub for ascending and cortical pathways (*Vinogradova, 2001*), and its output is conveyed to numerous brain regions, directly or from the next station in the CA1 after bilateral integration (*Swanson et al., 1981*). The left and right CA3 are connected reciprocally through the ventral hippocampal commissure (VHC), and they also send excitatory inputs to the CA1 on both sides of the brain through an associational (Schaffer)-commissural system that converges on pyramidal cells (PCs, *Figure 1A*). Hence, this system represents an ideal model to explore the flow of activity and its integration in bilateral networks. In addition, numerous electrophysiological studies show that the CA3 region behaves as a powerful gamma oscillator with inhibitory and excitatory local network components (*Traub et al., 1996*; *Fisahn et al., 1998*; *Bartos et al., 2002*; *Csicsvari et al., 2003*; *Hájos et al., 2004*; *Mann et al., 2005*; *Oren et al., 2006*; *Hájos and Paulsen, 2009*; *Pietersen et al., 2009*; *Gulyás et al., 2010*; *Mann and Mody, 2010*; *Kohus et al., 2016*). Whether and how the left and right CA3 gamma oscillators are coupled and how this influences the bilateral activity and performance of the hippocampus have not been explored.

We set out to compare the dynamic relationships of two network components that can be addressed on a cell-assembly basis using pathway-specific LFPs, the CA3 PC excitatory output reflected in CA1 Schaffer LFPs, and a somatic inhibitory input to the same population ($CA3_{som}$). We also studied the impact of excitatory bilateral convergence on the output of neurons in the next station, the CA1, by comparing the waveform details of Schaffer gamma waves on both sides and cell firing of CA1 units. To extract the Schaffer and $CA3_{som}$ LFP activities from the field contributions

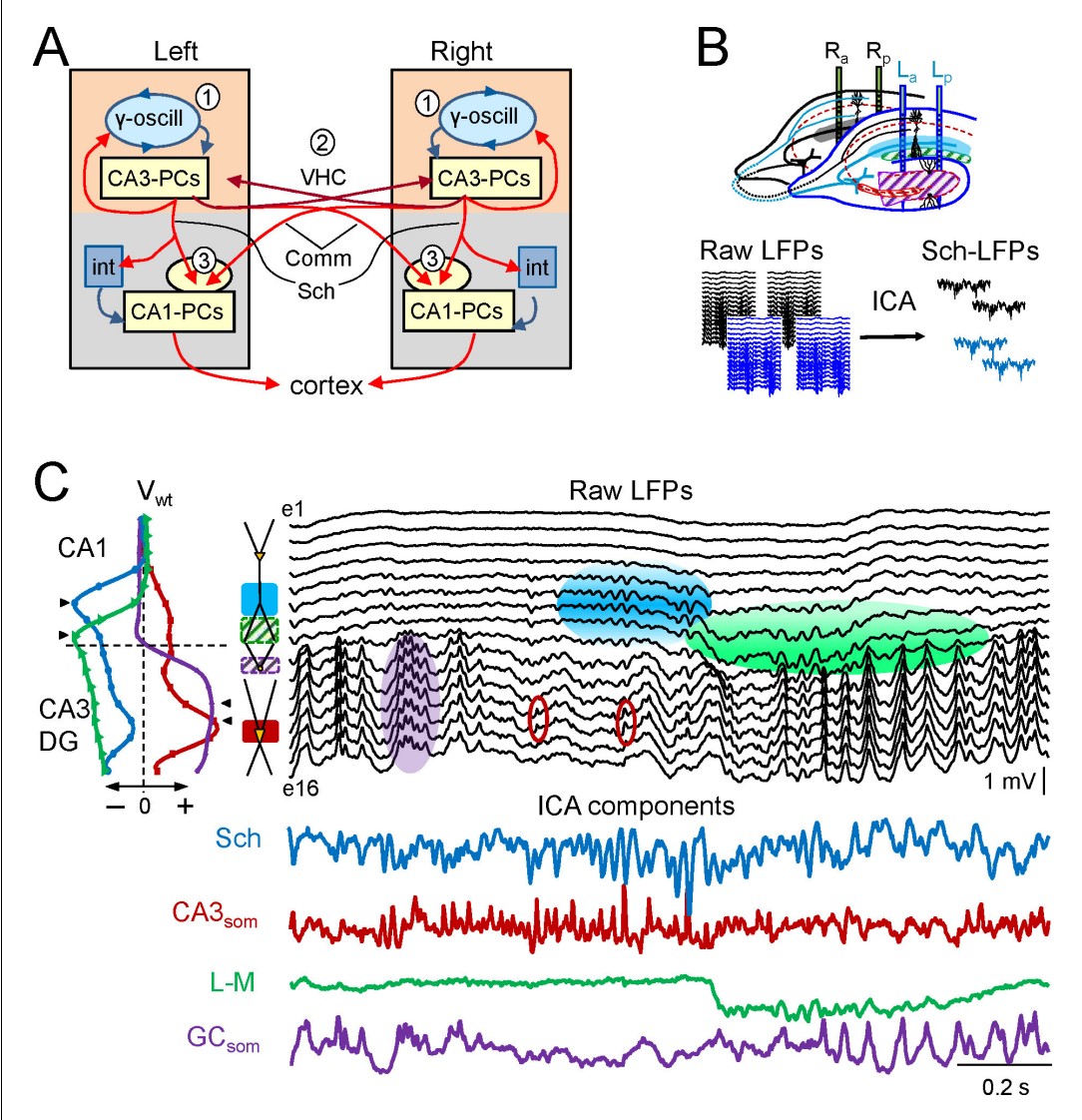

**Figure 1.** Experimental paradigm and clean out of the Schaffer and CA3som activities. (**A**) Functional characteristics of the bilateral CA3→CA1 segment: (1) an intrinsic gamma oscillator fueled by inhibition in each CA3 region produces gamma output from PCs; (2) The left and right CA3-PCs are interconnected through the ventral hippocampal commissure (VHC, maroon arrows), enabling the coupling of CA3 gamma oscillators; (3) The excitatory outputs of CA3-PCs from both sides converge in each CA1 (Schaffer and Commissural pathways). (**B**) Experimental setup. Two-shank linear arrays were located at homotopic sites of the dorsal left and right hippocampi. Recordings were acquired simultaneously and each group was analyzed separately by an Independent Component Analysis (ICA). (**C**) ICA of a sample epoch across the CA1 and CA3/DG layers. In raw LFPs (black traces), several bands of coherent voltage fluctuations are observed that indicate multiple activation in different synaptic territories (three are outlined by filled ovals spanning the CA1 and the Dentate subfields, while small maroon ovals mark activity in the st. pyramidale of the CA3). The ICA returns the spatially-coherent components and provides readout of the temporal dynamics free of a contribution by the others. A set of components or LFP-generators was obtained per shank, each with a characteristic spatial distribution or voltage weight ($V_{wt}$) that enabled matching between shanks. Details of the extraction are in *Figure 1—figure supplement 1*. Colored traces from top to bottom: Schaffer (cyan), CA3$_{som}$ (maroon), lacunosum-moleculare (green), and GC$_{som}$ (purple). The amplitudes are normalized (0.37:0.25:0.84:1). In other figures voltage units are employed that were estimated for the sites with maximum power (triangles in $V_{wt}$ plots).

The following figure supplement is available for figure 1:

**Figure supplement 1.** Details of the extraction and ICA performance.

added by other concurrent pathways, we applied a spatial discrimination method to multisite linear LFP recordings (*Figure 1B*) (*Herreras et al., 2015*). This approach provides unprecedented temporal precision and specification of the origin such that each spatially-isolated activity reflects the envelope of postsynaptic currents elicited by an afferent population on another (*Makarova et al., 2011*; *Fernandez-Ruiz et al., 2012a*; *Martín-Vázquez et al., 2013*; *Schomburg et al., 2014*; *Herreras et al., 2015*). Indeed, we previously reported that the Schaffer LFPs contain a regular succession of gamma waves (gamma strings) that reflect excitatory packages of extremely variable amplitude (*Fernandez-Ruiz et al., 2012a*, *2012b*), these being elicited in CA1 PCs by the synchronous firing of CA3 PC assemblies at gamma intervals (*Fisahn et al., 1998*; *Hájos and Paulsen, 2009*; *Takahashi et al., 2010*; *Fernandez-Ruiz et al., 2012a*). The respective excitatory and inhibitory nature of Schaffer and $CA3_{som}$ gamma waves has also been established previously (*Hájos et al., 2004*; *Benito et al., 2014*).

These studies indicate that Schaffer and $CA3_{som}$ gamma waves are generally larger and lead from the right lobe, while the $CA3_{som}$ component forms part of a local gamma oscillator that paces outgoing excitatory Schaffer packages. These form strings of gamma waves that may run unilaterally as seen upon VHC disconnection. In both sides, Schaffer waves may lead at different antero-posterior CA1 locations indistinctly, although they are all submitted to global bilateral asymmetric entrainment that optimizes bilateral convergence in the CA1. This inference is supported by the preferred firing of CA1 pyramidal cells during bilateral waves in contrast to the preferred unilateral driving of putative interneurons.

## Results

The mean frequency of LFPs recorded in the stratum radiatum of CA1 was $45.2 \pm 1.2$ Hz (estimated from the autocorrelation function on 10 min epochs, n = 7 animals), which corresponds to the so-called low-gamma frequency band that characterizes the hippocampal segment under study (*Csicsvari et al., 2003*; *Fernandez-Ruiz et al., 2012a*, *2012b*; *Martín-Vázquez et al., 2013*; *Schomburg et al., 2014*). Here we use the term gamma wave to refer to each pulse-like fluctuation of the LFP repeating at intervals of 20–25 ms, i.e. the wavelength of the oscillation, regardless of the duration of individual waves that differ for specific synaptic pathways (see below). Epochs containing layer-specific strings of gamma waves were observed in all hippocampal subregions, but they blend with activities at multiple frequencies and temporal patterns in a variable fashion (*Figure 1C*). Accordingly, the precise time course of individual waves cannot be reasonably assigned to a specific synaptic pathway. By using independent component analysis (ICA) (*Bell and Sejnowski, 1995*) to maximize spatially coherent activity, we isolated a small number of pathway-specific LFPs (colored traces) with characteristic spatial voltage weights ($V_{wt}$) that corresponded to a linear sample of the voltage shell produced by each synaptic pathway in the volume (*Herreras et al., 2015*). The populations of origin and the postsynaptic targets (pyramidal or granule cells) have been identified elsewhere (*Bell and Sejnowski, 1995*; *Korovaichuk et al., 2010*; *Fernandez-Ruiz et al., 2012a*; *2012b*; *Martín-Vázquez et al., 2013*, *2015*; *Schomburg et al., 2014*). We here studied the Schaffer-specific CA3→CA1 excitatory input that reflects assembly organization of CA3 output, and the somatic CA3 inhibitory input (*Figure 1C*, cyan and maroon traces, respectively). Details of the contributions cleaned of any influence of nearby sources in the stratum lacunosum-moleculare (s.l-m.) of the CA1, or the Dentate Gyrus (DG), respectively, are shown in *Figure 1—figure supplement 1*.

### Schaffer gamma activity is stronger and precedes in the right side

The Schaffer activity contained strings of gamma waves, irregular fluctuations, and Sharp-Waves (SPW), each reflecting different regimes of organized firing in CA3 PCs (*Benito et al., 2014*). The occurrence and duration of gamma strings was unpredictable, ranging from a few waves to many seconds and they were no further characterized. To test the bilateral complementariness we first compared the time course of Schaffer activity as a whole (containing interspersed gamma strings and irregular fluctuations) in two pairs of left-right homotopic sites of the dorsal hippocampus (*Figure 2A* and *Figure 2—figure supplement 1*). Epochs with frequent SPWs were avoided or these were removed as they contribute disproportionately to the statistics. We found significant wide band coherence between pairs of sites that are 0.5 mm apart within antero-posterior (a-p) lamellar

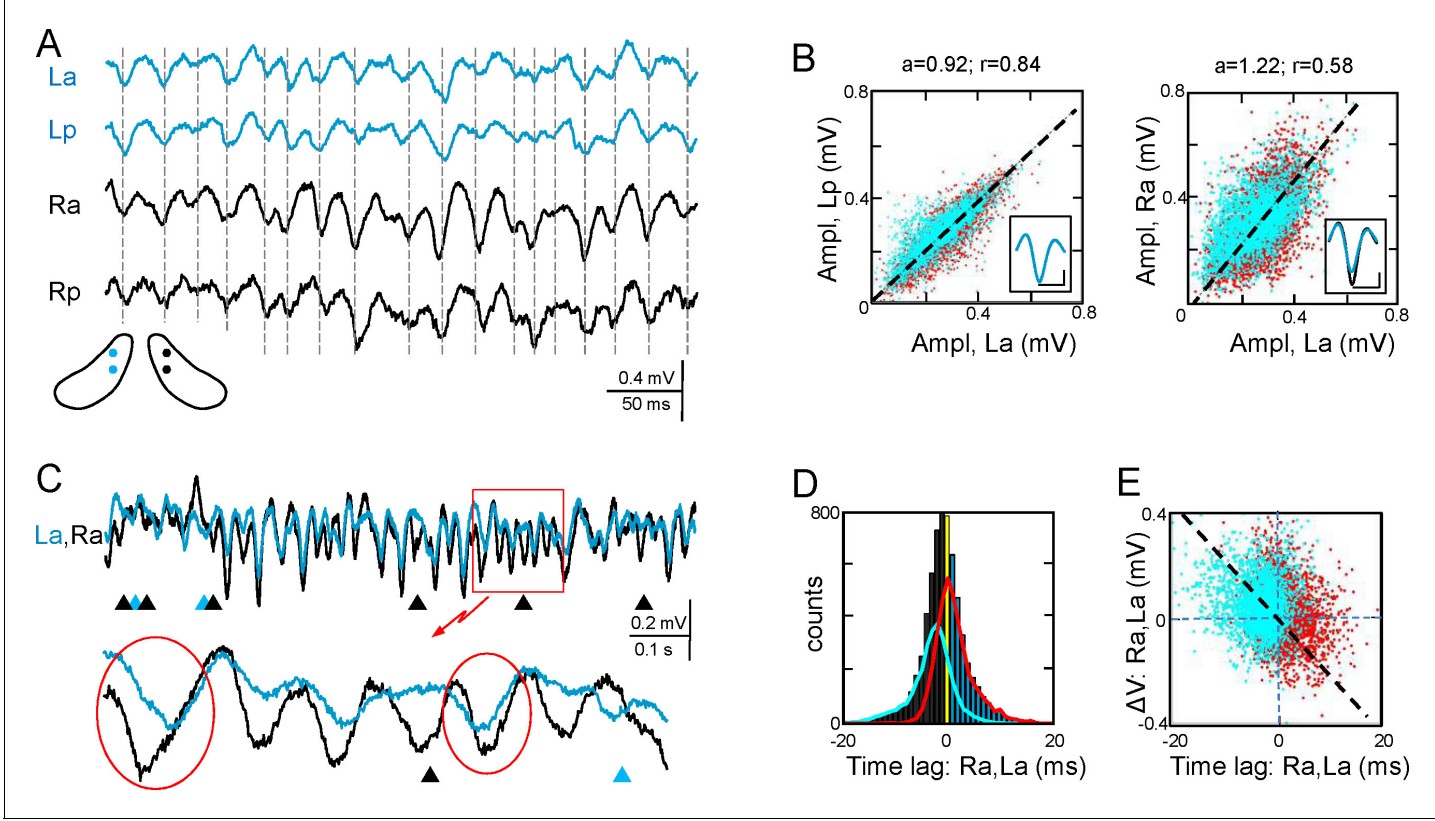

**Figure 2.** Functional asymmetry in the bilateral CA3-CA1 system. (**A**) Sample string of Schaffer-gamma obtained from four sites. Individual waves coincide regardless of their amplitude. Globally, Schaffer-gamma is larger on the right side. The scheme shows the location of recordings from a coronal view (*Figure 2—figure supplement 1*). (**B–E**) Representative experiment showing the features of individual waves compared pairwise within (La, Lp) and between hippocampi (La,Ra) (n = 6623 pairs of waves in 167 s). The blue and red dots belong to the pairs when L or R waves were longer, respectively. The population statistics and additional examples are in *Figure 2—figure supplements 2–3*. (**B**) Waves co-vary closely in the same side (left) and much less so between sides (right): *a*, best fit tangent; *r*, correlation coefficient. The insets show superimposed averaged waves (cal: 20 ms and 100 µV). (**C**) A string of Schaffer gamma shows unilateral waves in both sides (triangles). In paired bilateral waves, either side may lead (ovals). (**D**) Bilateral synchronization was measured from the start of the waves (time lag). The positive and negative values indicate that L or R waves led, respectively. R waves preceded more often (black bars), the bilateral lag being larger when R-waves were longer (line subplot in blue). (**E**) The amplitude difference between paired waves in the right and left sides is plotted against their time lag. Larger waves on any side had a tendency to lead.

The following source data and figure supplements are available for figure 2:

**Source data 1.** Spreadsheet containing measurements of the LFP generators and extracted waves for each experiment.

**Source data 2.** Schaffer LFP generators and extracted waves for the experiments used in *Figures 2 and 4*.

**Figure supplement 1.** Histological and electrophysiological localization of recording sites.

**Figure supplement 2.** Additional data and population statistics for the comparison of intra and interhippocampal CA3→CA1 Schaffer activity.

**Figure supplement 3.** Additional sample traces of Schaffer activity simultaneously obtained from CA1 homotopic sites.

strips. This coherence is reduced and restricted to the gamma band in bilateral comparisons (*Figure 2—figure supplement 2*). The cross-correlation coefficient (CC) behaved similarly (La-Lp, 0.84 ± 0.03; Ra-La, 0.57 ± 0.04; $F_{(1,12)}$ = 26.9; p<0.001; mean of epochs lasting 85–167 s, n = 7). The mean $\tau_{max}$ of CCs was similar for a-p sites (0.32 ± 0.2 ms) but shifted -0.82 ± 0.3 ms ($F_{(1,12)}$ = 8.5, p=0.01) for La-Ra comparisons, with the right side leading.

Gamma strings of tight left-right amplitude co-modulation between paired waves (*Figure 2A and C*) alternate with others of loose covariation (*Figure 2—figure supplement 3*). Therefore, we quantified the features of individual waves obtained through a deconvolution approach (see *Figure 2— source data 1 and 2*, and Materials and methods) and compared these between different sites. To avoid noisy LFP events, waves were only considered when they lasted >5 ms and reached >20 µV. The amplitude, duration and start time of individual Schaffer gamma waves reflect the size and synchronization of CA3 PC assemblies. Waves were designated as paired when they overlapped at two sites by at least 70% of their duration. Despite of a clear amplitude fluctuation at any site, paired waves were nearly identical along lamellar strips in one side (*Figure 2A and B*; covariance was $\rho_A$ = 0.78 ± 0.02 for the amplitude and $\rho_D$ = 0.63 ± 0.02 for the duration in La:Lp comparison), while they showed less co-variation between homotopic La-Ra sites ($\rho_A$ = 0.5 ± 0.04 and $\rho_D$ = 0.44 ± 0.03; see population statistics in *Figure 2—figure supplement 2B,C*). The matching waveforms and strong covariation of ipsilateral waves rule out the possibility that the modulation of amplitude over successive waves is due to a different site of origin of the waves and hence, the distance to the electrodes (*Benito et al., 2014*). Rather, it is consistent with a coarse lamellar-like distribution of Schaffer fibers in the CA1 for all waves, regardless of their amplitude. The mean wave duration was the same on both sides (R and L: 26.7 ± 0.7 ms; $F_{(1,12)}$ = 0, p=0.96).

We found notable bilateral asymmetries. In particular, waves on the right side were larger in amplitude (Ra: 299 ± 11 µV; La: 237 ± 15 µV; $F_{(1,12)}$ = 12, p=0.005, n = 7 animals: *Figure 2B–D*). Interestingly, bilateral waves were rarely initiated synchronously. Although either side may lead (ovals in *Figure 2C*), it was more common that waves in the right side did so (55.6 ± 0.8% *vs* 44.4 ± 0.8% ). On average, the R waves preceded with similar values between anterior or posterior sites (Ra-La: 0.62 ± 0.14 ms; Rp-Lp: 0.67 ± 0.18 ms). We also found a notable proportion of unilateral waves (*Figure 2C*), with a higher incidence on the right side (R: 11.7 ± 2.4%; L: 5.5 ± 1%). These unilateral waves were smaller than bilateral ones in the right hemisphere (212 ± 20 µV; bilateral *vs.* unilateral: p=0.002), but not in the left one (192 ± 20 µV; p=0.09).

To countercheck the overall primacy of Schaffer activity on one side we applied the Granger Causality (GC) test between the right and left Schaffer generators over 90 s epochs. *Figure 3A* shows R and L Schaffer activations in a representative experiment. The GC test confirmed that there are statistically significant and reciprocal relations between the R and L sides (*Figure 3B*: p=$10^{-6}$ and p=$10^{-8}$ for L to R and R to L relations, respectively). Moreover, the functional relation peaked in the low-gamma frequency band (30–50 Hz) and the right to left link was strongly dominant over time (*Figure 3C,D*). This result was verified for all seven animals.

Although the GC test confirmed a preferred right to left directionality, it provided no time lag between the Schaffer activities. We crosschecked the lag obtained from the start times of the paired gamma waves using an additional test to evaluate the phase shift between the Schaffer activities. *Figure 3E* shows the histogram of the phases in Schaffer activity in the left side with onsets related to zero phases in the right side (corresponding to the beginning of LFP events in that side). The circular statistics confirmed the presence of a highly significant peak at $\Delta\phi$ = 0.22 ± 0.04 radians. Thus, the right side indeed appears to lead the generation of activity in Schaffer pathways. By evaluating the mean frequency of the generators (~45 Hz) we can estimate the corresponding time lag $\Delta t$ = 0.95 ± 0.2 ms, in a good agreement with the value obtained by the analysis of paired Schaffer waves (1.12 ms for this experiment). Again this result was verified in all seven animals ($\Delta\phi$ = 0.12 ± 0.06 radians; $\Delta t$ = 0.54 ± 0.26 ms), with the mean time lag being roughly equal to the $\tau_{max}$ in the CCs.

## Variable ipsilateral dynamics subjects to global asymmetric bilateral coupling

Since the duration of paired waves was not identical, we explored whether the leading site had any relation to the wave's features. The more relevant results were obtained when paired left-right waves were sorted by the site at which they showed longer duration (L or R, anterior or posterior) (*Figure 2B,D,E*). Notably, the right-left lag was accentuated when R-waves were longer (a, p: 3.09 ± 0.06 and 3.03 ± 0.17 ms), whilst longer L-waves also led to righ-hand ones albeit by a smaller amount (a, p: 1.97 ± 0.13 and 1.91 ± 0.23 ms, n = 7, $F_{(1,12)}$ = 517, p<0.001) (*Figure 2D*, and *Figure 2—figure supplement 2*). The data from different individuals indicated that these lags were robust for different antero-posterior locations of the shanks or when there was a slight a-p displacement between the right and left sides (*Figure 2—source data 1* and *Figure 2—figure supplement 1*). We also

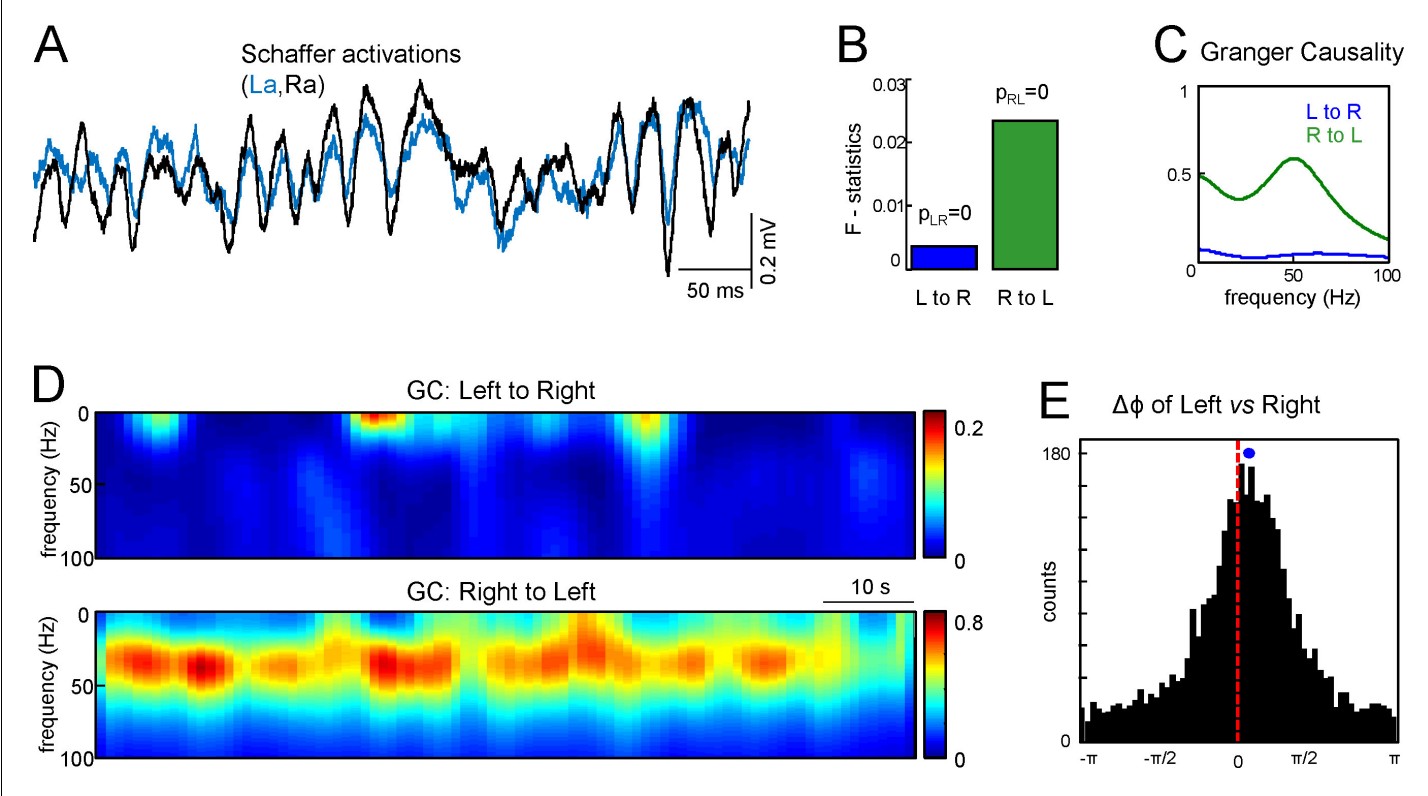

**Figure 3.** Assessment of functional asymmetry with Granger causality and phase relations. (**A**) A short epoch of activations of the right and left Schaffer pathways. (**B**) F-statistics for Granger Causality (GC) test revealing significant reciprocal influence from R to L and from L to R sides. (**C**) Frequency dependence of GC. R to L relation exhibits a peak at gamma frequency. (**D**) Time-frequency display of the GC index. R to L relationship is stronger and more persistent. (**E**) Distribution of phases in the L side with onsets related to zero phases in the R side, i.e., when field events begin. The mean phase lag of 0.22 rad (corresponding to 0.95 ms time lag) is highly significant. The population data is indicated in the text.

found a tendency towards a lead by longer and larger waves irrespective of the leading side (*Figure 2E*). These observations indicated an independent discharge by left and right CA3 assemblies that project to the anterior or posterior sites of CA1 indistinctly, while the overall lag towards the right-hand side supports a global right-to-left asymmetric entrainment. Such directionality is also supported by the fact that the mean lag from crossed $L_p$ to $R_a$ sites (1.13 ± 0.25 ms) was roughly equal to the accumulated lag from right to left plus the lag used by a-p axonal conduction: $R_a \rightarrow L_a$ (0.62 ± 0.14 ms), $L_a \rightarrow L_p$ (0.52 ± 0.1 ms). In turn, the $L_a$ to $R_p$ lag became balanced (–0.1 ± 0.24 ms) as expected if the right-to-left lag were absorbed by the opposite sign of the a-p lag: $L_a \rightarrow R_a$ (-0.62 ± 0.14 ms), $R_a \rightarrow R_p$ (0.67 ± 0.33 ms).

The preferred lead of longer/larger Schaffer gamma waves in either side might suggest that CA3 clusters on one side excite CA3 clusters on the other, which then set up delayed contralateral Schaffer waves in the CA1. We explored this possibility by estimating the minimum time used by CA3 spikes on one side to get to CA3 PCs through the latency of the contralateral CA3 antidromic population spike elicited from the septal pole of the Left CA3, which was 2.5 ± 0.1 ms (n = 6). We also estimated the shortest possible lag for the direct driving of CA3 to the contralateral CA1 through the latency of the evoked commissural fEPSP in CA1, which amounted 5.8 ± 0.3 ms (n = 7, i. e.: three-fold the L-R lag for spontaneous paired waves). These results leave some room for a reduced fraction of the bilateral waves to be explained by the inter-hippocampal excitatory driving of unilaterally ignited CA3 clusters, although the presence of numerous unilateral waves points to additional mechanisms.

Notably, side-independent instigation of Schaffer waves was also found in intra-hippocampal a-p comparisons. Thus, while on average Schaffer waves led in anterior sites (0.52 ± 0.06 and 0.67 ± 0.33

ms for the L and R sides, respectively; L-R: $F_{(1,12)} = 0.3$, p=0.6) in compliance with a global antero-posterior lamellar topology of Schaffer fibers, we found quite different lags depending on which a-p site had longer waves. For waves that were longer in anterior sites the a-p lag increased to $2.6 \pm 0.3$ and $2.2 \pm 0.3$ ms in the right and left sides, respectively (R-L: $F_{(1,12)} = 0.65$, p=0.4; total *vs.* longer R: $F_{(1,12)} = 102$, p<0.001). More notably, when posterior waves were longer they preceded anterior ones by $1.13 \pm 0.5$ and $1.8 \pm 0.3$ ms for the R and L sides, respectively (R-L: $F_{(1,12)} = 1.24$, p=0.29; total vs. longer L: $F_{(1,12)} = 53$, p<0.001). Thus the large CA3 PC assemblies initiate firing on any side and location, regardless of the overall bilateral coupling that maintains a global precedence on the right-hand side.

## Tight ipsilateral synchrony but weak bilateral entrainment characterizes the CA3 gamma waves in the soma layer

To delve further into the mechanisms of bilateral entrainment of Schaffer gamma oscillations, we explored the ipsi- and bilateral expression of the gamma activity in the soma layer of the CA3 region itself, and its relationship to Schaffer activity (outgoing CA3 quanta). This so-called the $CA3_{som}$ generator is known to be part of a gamma pacemaker in this region (*Traub et al., 1996*; *Fisahn et al., 1998*; *Bartos et al., 2002*; *Csicsvari et al., 2003*; *Hájos et al., 2004*; *Mann et al., 2005*; *Oren et al., 2006*; *Hájos and Paulsen, 2009*; *Pietersen et al., 2009*; *Gulyás et al., 2010*; *Mann and Mody, 2010*; *Kohus et al., 2016*). The $CA3_{som}$ gamma-paced wavelets were recorded over 1 to 3 contiguous electrodes in different animals and they appeared as positive non-overlapping events riding on a flat baseline (*Figure 1C* and *Figure 1—figure supplement 1B*).

Ipsilateral comparisons of $CA3_{som}$ LFPs between anterior and posterior sites in the two sides were only possible in two animals (see *Figure 4* for an illustrative experiment), although partial comparisons were obtained in another four. The data were pooled by side (La-Lp and Ra-Rp: n = 7) or position (La-Ra and Lp-Rp: n = 8), and like the Sch LFPs, the $CA3_{som}$ LFPs displayed close-fitting within-side activities, whereas bilateral synchrony was far less marked and mismatches of individual waves were more frequent (unilateral $CA3_{som}$ waves: $22 \pm 1.2$ and $26 \pm 2.3\%$ for the L and R sides, respectively; *Figure 4A*, La-Lp vs. Lp-Rp traces). Accordingly, the antero-posterior wide band coherence gave significant values for frequencies above 25 Hz in all cases, whilst none gave significant values for bilateral comparisons (*Figure 4A*, spectral coherence plots). The CCs behaved similarly (a-p: $0.66 \pm 0.6$; L-R: $0.27 \pm 0.02$; $F_{(1,12)} = 32$, p<0.001, with a non-significant phase difference between a-p comparisons ($\tau_{max}$: $0.1 \pm 0.2$ ms), but similar to Schaffer activity in R-L comparisons ($0.42 \pm 0.8$ ms). The comparisons between extracted $CA3_{som}$ gamma waves yielded similar results as that for Schaffer activity (*Figure 4—figure supplement 1A*). Thus, the amplitude covariance of bilateral waves was stronger in intra ($\rho A = 0.62 \pm 0.06$) than interhippocampal waves ($\rho A = 0.24 \pm 0.04$; $F_{(1,14)} = 42$, p<0.001), although the amplitude did not differ in the left and right hemispheres (La: $162 \pm 26$; Ra: $183 \pm 28$ μV; ($F_{(1,10)} = 0.32$, p=0.58). Also, on average R-waves led by $0.34 \pm 0.1$ ms (n = 8), and this lag increased to $2.72 \pm 0.01$ ms when the R-waves were longer (total vs. longer R: $F_{(1,14)} = 297$, p<0.001), and longer L-waves also preceded the right-hand ones by $2.28 \pm 0.02$ ms (total vs. longer L: $F_{(1,14)} = 643$, p<0.001; (*Figure 2—source data 1*), matching the relationships to the Schaffer waves.

We then related the $CA3_{som}$ and Schaffer activities and over a large time-scale (seconds), the two followed similar trends (*Figure 4—figure supplement 1B*) by displaying parallel gamma strings and pauses. However, clear differences were observed over shorter time-scales (hundredths of milliseconds). Since we observed no differences between the sides and sites, we pooled the data from all shanks and animals where both activities could be recovered (n = 19). As expected from the flat baseline in the $CA3_{som}$ component, in all cases the spectral coherence between $CA3_{som}$ and Schaffer activities was only significant in the gamma band (*Figure 4B*). Despite the tight temporal relationship between individual Schaffer and $CA3_{som}$ waves, the CC was only moderate ($0.4 \pm 0.02$, $\tau_{max}$ of $4.72 \pm 0.4$ ms), in part due to the different duration of the gamma waves (Schaffer waves are ~2.5 times longer than the $CA3_{som}$ ones and hence, the respective time courses are largely out-phased).

We also applied the GC test to explore the relationship between the $CA3_{som}$ and Schaffer activities obtained at the same a-p site and side (two shanks on each of six animals: n = 12). In all cases, the GC test indicated statistically significant reciprocal relationship between the two activities that peaked in the gamma frequency band, and the $CA3_{som}$ to Schaffer link strongly dominated. We also evaluated the phase shift that yielded a highly significant peak at $\Delta\phi = 1.46 \pm 0.18$ radians, the

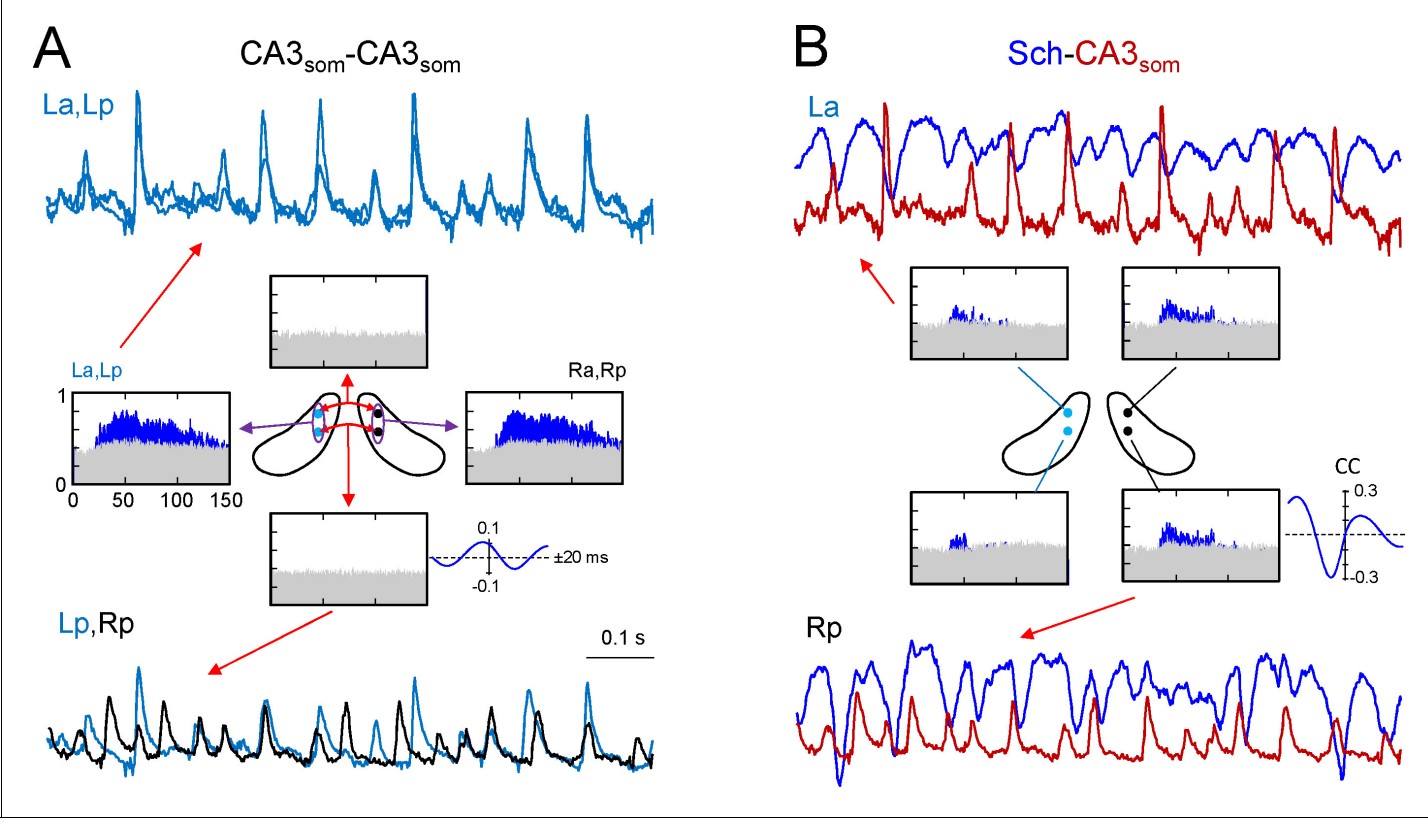

**Figure 4.** CA3som gamma activity has weak bilateral coherence but is coupled to ipsilateral Schaffer. (**A**) Comparison of CA3som activities between pairs of sites. The histograms of spectral coherence only show significant values (blue) for ipsilateral comparisons. The sample traces show tight matching in superimposed activities at a-p sites (upper traces) and frequent mismatch in bilateral comparisons in the same epoch (lower traces). Cyan and black traces correspond to the left and right sides. (**B**) Comparisons between Schaffer and CA3som activities (blue and maroon traces, respectively). The spectral coherence showed significant values only at gamma frequency at all four sites. Sample traces show strong wave-to-wave coupling despite the poor amplitude covariation. The CC was strong and showed a marked left-shift that mostly originates from the different waveform of individual waves. All data were taken from the same animal (see population statistics in the text, and additional analyses in *Figure 4—figure supplement 1*).

The following figure supplement is available for figure 4:

**Figure supplement 1.** Additional features of the CA3som activity and waves.

CA3som leading Schaffer activity. The estimate time lag was $\Delta t = 6.18 \pm 0.7$ ms, which is in a good agreement with the value of the $\tau_{max}$ obtained from CCs.

With regards to the features of extracted waves, the most relevant finding was the poor amplitude co-variation between paired Schaffer and CA3som gamma waves recorded by the same shank ($\rho A = 0.16 \pm 0.02$). This observation may indicate a different topology, composition, or location of the CA3 PCs receiving a synchronous inhibitory volley and those setting the Schaffer waves in the CA1. In addition, the time lag between paired Schaffer and CA3som waves was not significant ($0.12 \pm 0.3$ ms), which should be interpreted in light of the different target cells and the reduced sampling ($42.3 \pm 2\%$ of CA3som waves remained unpaired). Globally, these results conform to a strong but not necessarily causal lock of oscillatory somatic inhibition in CA3 to the outgoing gamma excitation, while the weak bilateral coupling of the former indicates that it operates mostly ipsilaterally.

## Bilateral Schaffer gamma waves promote firing in pyramidal cells while putative interneurons prefer a unilateral input

Schaffer gamma waves are excitatory inputs to CA1 units and they arrive synchronously with those from the contralateral side due to the tight coupling of CA3 gamma oscillators, which should have an impact on their output. We checked this inference by exploring the combined efficiency of

bilateral gamma waves to bring CA1 units to fire. The excitatory nature of Schaffer waves was confirmed by the stronger response of CA1-PCs to bilateral as opposed to unilateral waves (*Figure 5A*, *Figure 5—source data 1*, and *Figure 5—figure supplement 1*). The precise timing of firing was not solely determined by these inputs since spikes were relatively dispersed over the gamma cycle (*Figure 5B* and *Figure 5—source data 2*). Nevertheless, each PC had preferred intervals from the start of paired waves, indicating that they were cell-specific optimal intervals for bilateral summation (*Figure 5—figure supplement 1*). In turn, a large proportion of putative CA1 interneurons showed tight phase-locked firing shortly after the start of the bilateral gamma waves, and they also fired more frequently than PCs in response to unilateral waves on the same side (*Figure 5B* and *Figure 5—figure supplement 1*).

Since unilateral Schaffer waves were slightly smaller than bilateral waves, we explored the possibility that increased association of PC firing to bilateral waves was due to stronger ipsilateral input rather than to the bilateral convergence of Schaffer and Commissural inputs. Given the sparseness of cell firing compared to the abundance and amplitude range of gamma waves we devised a global approach by estimating the probability with which spikes emitted by PCs deviate from spikes drawn from a random spike train over the instantaneous power distribution of the Schaffer generator. The typical distribution of the instantaneous power of the Schaffer generator (*Figure 5—figure supplement 2A*) has two regions that follow power laws, $ax^k$, with different scaling constants ($k_1 = -1.22$ for power < 1, and $k_2 = -2.66$ for power > 1; to compare between different experiments we scaled the generator power equal to 1 at the critical point). The first region corresponds to noisy dynamics, whereas the second one is associated with rarer but sufficiently strong gamma waves (bi- and unilateral). The mean differential probabilities of PC spikes emitted during unilateral R and L waves (blue and red curves, respectively) or bilateral waves (black curve; data obtained over n = 40 PCs) were plotted (*Figure 5—figure supplement 2B*). In both cases the differential probabilities were negative in the noisy regions (power < 1), i.e.: PC firing was unlikely in association with small noisy events. For strong gamma waves (power > 1) bilateral spikes exhibited quite a flat distribution over the wave power. Moreover, the probability was smaller than that for spikes emitted during unilateral gamma waves on the same side. Thus, the coincidence of left-hand and right-hand waves (i.e.: bilateral excitation) plays crucial role in PC firing, while the associated power increase has a marginal effect.

## Disruption of the interhippocampal connection reveals lateralized gamma strings

The presence of unilateral waves, frequent epochs of loose bilateral co-modulation, and the wide range of delays between bilateral waves appears to be incompatible with their joint initiation through the ipsi- and contralateral axonal branches of PCs in a CA3 assembly on the leading side. Thus, we explored the role of interhippocampal CA3←→CA3 connections by micro-injection of lidocaine in the VHC. Effective disruption was confirmed by the abolition of evoked fEPSPs in the CA1 of both hemispheres following stimulation through the injecting pipette (*Figure 6A,C*). Notably, although Schaffer gamma strings were still observed bilaterally, frequent unilateral strings occurred that were not seen in the controls (*Figure 6B* and *Figure 6—figure supplement 1*). When they co-occurred we found that the overall power and wave characteristics on either side did not change (*Figure 6C*, *Figure 6—source data 1*, and *Figure 6—figure supplement 1*), evidence that each CA3 can autonomously generate gamma in vivo, as reported earlier for chemically-induced gamma oscillations in vitro (*Oren et al., 2010*). Coarse bilateral co-modulation was maintained, albeit reduced (CC, 0.62 ± 0.03 *vs* 0.45 ± 0.05; p=0.007; mean of epochs lasting 100 s in n = 4 animals; *Figure 6D and E*). Moreover, bilateral gamma coherence was lost (*Figure 6F* and *Figure 6—source data 1*) and consequently, the L-R synchrony of individual gamma waves disappeared while the amplitude variability of the waves remained high in both sides (*Figure 6C*). This indicates that different gamma strings may circulate independently through L and R hippocampi.

It might be expected that bilateral disruption affects the firing of units receiving ipsi and contralateral CA3 inputs. Given their low firing rate, PCs require long recording times and thus, we used an alternative experimental design that offers higher long-term stability, as the blockade of the CA3 itself on the left side. Units in the right CA3 and CA1 subregions were recorded for 700 s before and after lidocaine administration. A total of 33 PCs and 23 putative interneurons were recorded (n = 4 experiments), and nearly all cells underwent changes in the mean firing rate following contralateral CA3 blockade (*Figure 6—figure supplement 2*). The firing of 10 out of 19 PCs in the CA1, and

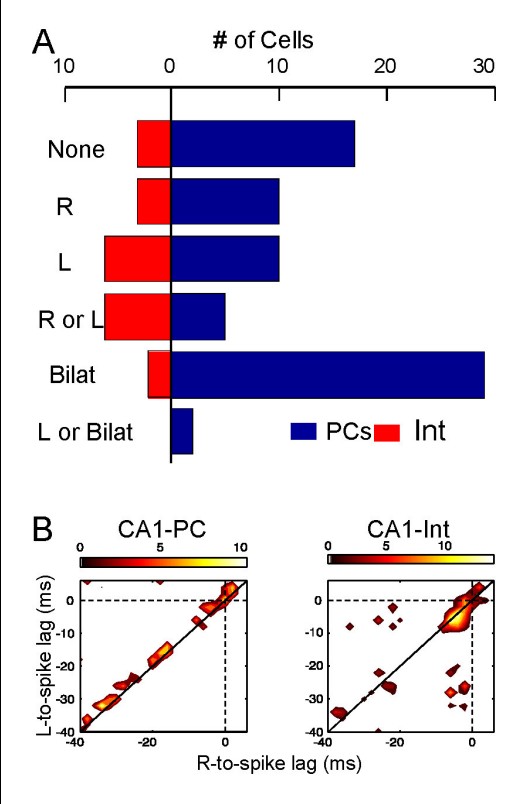

**Figure 5.** Left-Right synchrony of Schaffer gamma waves drives firing of CA1 PCs and putative interneurons distinctly. (**A**) Spike trains were correlated to the activity of the Schaffer generators in both sides, and the spikes were associated to unilateral (only R, only L, R or L groups) or bilateral (i.e. synchronous) gamma waves, or to none. Only statistically significant cells are represented. PCs fired preferentially upon convergence of bilateral gamma waves, while putative interneurons showed a marked unilateral drive. (**B**) Firing of representative cells in function of when the gamma waves started in the L and R sites (additional data in *Figure 5—figure supplement 1*). Spikes were sorted that occurred within 40 ms of the beginning of a gamma wave in both sides (paired waves). The spike occurs at (0,0), while the X and Y axes represent the start time of the R and L gamma waves, respectively. The calibration bar indicates the firing density upon removal of chance firing. Most firings approach the diagonal as a result of tight bilateral gamma coupling. However, PCs fired at varying cell-specific intervals, while firing of putative interneurons was phase-locked with a short lag from the start of the excitatory gamma waves. The relation of spike firing to LFP power is shown in *Figure 5—figure supplement 2*.

The following source data and figure supplements are available for figure 5:

**Source data 1.** Spreadsheet containing the unitary results for the bilateral/unilateral test performed on CA1 units in *Figure 5A*.

*Figure 5 continued on next page*

of 4 out of 14 PCs (all from the same experiment) in the CA3 increased. As for interneurons, 7 out of 9 in the CA1 and 8 out 14 in the CA3 reduced firing. Neither the overall power nor the wave characteristics changed on the side contralateral to the lidocaine injection.

## Discussion

To date, bilateral networks have only been accessible from a macroscopic perspective. Thus, we sought here to obtain mechanistic evidence for the bilateral integration of spontaneous activities in the intact animal, focusing on those running through the CA3-CA1 segment of the two hippocampal lobes. By inspection of CA3-originated Schaffer potentials in the CA1, we established that the gamma-paced firing of CA3 PC assemblies is bilaterally entrained in an asymmetric manner, such that the right side commands over the left. Such dominance may serve to expedite and fine-tune a bilateral trade-off of autonomous oscillators on both sides of the hippocampus, as it would optimize the convergence of lateralized excitatory gamma strings on CA1 units. Indeed, CA1 PCs fire better upon bilateral gamma waves and although disruption of the VHC modifies their individual firing rate, the overall PC population output does not vary. These findings support the view that the two lobes of the hippocampus may convey different aspects of information that are cross-checked in the CA3-CA1 segment, although the output to cortical structures probably remains lateralized.

The observations are largely based on the examination of waveform details in paired gamma waves from ICA components of LFPs obtained at different sites. Gamma waves are present in multiple brain regions but their quantitative use as individual entities to represent the activity of cell-assemblies is hampered by the technical problems inherent to field potentials. In the hippocampus in particular, several sources of gamma activity have been reported in different strata and subregions (*Makarov et al., 2010*; *Schomburg et al., 2014*), and these contaminate each other through volume conduction (*Martín-Vázquez et al., 2013*; *Schomburg et al., 2014*). This makes the quantitative estimation of waveform parameters unreliable. However, when estimated on ICA-cleaned Schaffer waves, the onset, duration and amplitude, can be used as faithful correlates of the size and firing synchrony of afferent CA3 assemblies (*Makarova et al., 2011*; *Fernandez-Ruiz et al.,*

*Figure 5 continued*

**Source data 2.** Analysis of CA1 units in relation to the occurrence of gamma waves (co-modulograms).

**Figure supplement 1.** Relation of spike firing to bilateral gamma waves (additional data).

**Figure supplement 2.** Distribution of PC firing induced by bi- or unilateral gamma waves over the power of the Schaffer activity.

*2012a*; *2012b*; *2013*; *Benito et al., 2014*; *Schomburg et al., 2014*).

## The right side leads: mechanistic implications

The GC test indicates that Schaffer activity on any side may predict activity on the other, and the R-to-L influence is more robust. However, the persisting finding among different experiments of a lower *p* for the R-to-L statistics versus L-to-R cannot be used for qualifying unequivocally a preferred directionality. Similar caution should be taken to interpret the lags yielded by the $\tau_{max}$ from CCs and the phase-difference test, both pointing to a precedence of right-hand over left-hand activities. In a strict sense, the primacy of the right lobe is supported by evidence from the extracted waves, which can be considered discrete events independent from each other: (a) the mean lag between paired bilateral waves shows quantitative advance on the right-hand side (around 0.6–0.8 ms); (b) The mean amplitude of Schaffer waves was 30% larger on the right-hand side; (c) $CA3_{som}$ waves showed a similar advance on the right-hand side; (d) The cumulated lag in crossed comparisons, either summed (Lp-Ra) or balanced (La-Rp), in consonance with the partial lags between a-p and R-L sites. The advantage of such asymmetry attains functional meaning when we consider that lateralized Schaffer gamma strings converge on the CA1 units of each side on a wave-to-wave basis.

Examination of the relative timing of the waves at different sites sheds light on the mechanisms underlying the organization and dynamic behavior of natural CA3 assemblies (for a summary see *Figure 7*). Since the fate of a Schaffer wave is to excite CA1 units where it integrates with input from the contralateral side it is crucial to optimize the temporal overlap of the respective excitatory envelopes. Therefore, the short mean lag between paired bilateral waves indicates that the system is so finely adjusted (see functional implications below). However, in terms of connectivity it appears too small lag to consider unilateral CA3 assemblies driving Schaffer waves on the opposite side to be a general mechanism. Yet, since waves may lead on either side, the mean bilateral lag is to a large extent balanced. Subgroups of longer left or right waves do present longer mean lags (1.5 and 3.2 ms). Nevertheless, the minimum time required for CA3 spikes to reach contralateral CA3 PCs is 2.5 ms, which when added to the time required for synaptic integration and assembly recruitment is insufficient to explain the lag in the majority of paired waves. Such view might change depending on how PC assemblies are recruited. For instance if PC assemblies are recruited through the so-called leader PC cells (*Fujisawa et al., 2006*; *Wittner and Miles, 2007*), the driving factor between such cells may require shorter lags. On the other hand, a direct contralateral driving of gamma waves in the CA1 via commissural fibers is less likely for several reasons: (a) the minimum lag of commissural evoked potentials is already in the higher range of the lags between spontaneous bilateral waves; (b) the commissural pathway contributes little to field potentials due to anatomical and geometrical factors (*Martín-Vázquez et al., 2015*), making gamma waves of such origin unlikely; (c) the presence of unilateral waves coincides on the same idea, i.e., since a CA3 assembly may initiate a Schaffer wave that has no contralateral counterpart, when contralateral waves appear they most likely have an ipsilateral origin; (d) finally, the disruption of the VHC, or the unilateral inactivation of the CA3 would markedly reduce the presence of Schaffer waves and dampen the mean power of the Schaffer generator, which does not happen. Therefore, although unilateral CA3 assemblies driving contralateral ones cannot be ruled out entirely, the most likely origin is an independent ipsilateral driving.

Schaffer waves may lead in anterior or posterior sites on the same side, and their mean lag increased far longer than expected for axonal conduction when longer waves led. Such long delays may indicate independent ignition of CA3 assemblies whose projection to CA1 may follow either lamellar or longitudinal topologies (*Li et al., 1994*), while the lead on anterior or posterior sites appears to reproduce the different septotemporal CA1 projections of the CA3 subzones (*Ishizuka et al., 1990*). We also reported previously that Schaffer coherence decreases rapidly over the septo-temporal axis (*Benito et al., 2014*). The emerging view (*Figure 7*) is that once every gamma cycle a collection of spatial modules of the CA1 dendritic space is activated over the septo-

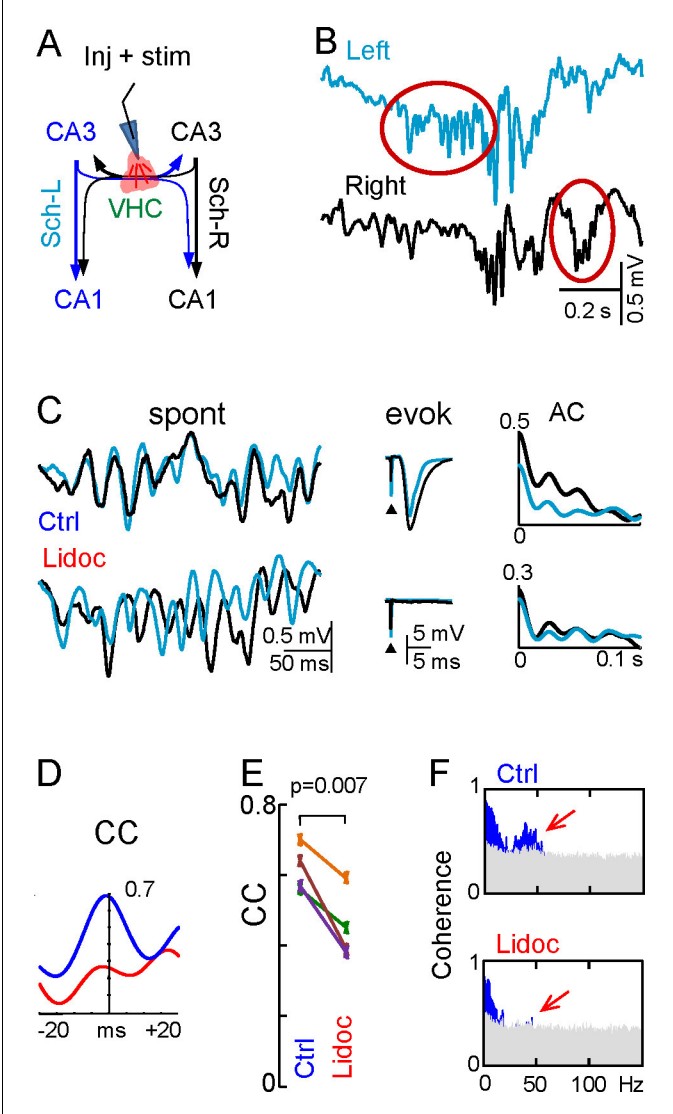

**Figure 6.** Disruption of the VHC uncouples CA3 gamma oscillators. (**A**) Experimental setup: lidocaine was injected through a pipette into the VHC, which also served to deliver electric pulses; (**B**) Gamma strings may appear on only one side after lidocaine injection (ovals) (additional examples in *Figure 6—figure supplement 1*). (**C**) Lidocaine in the VHC modifies L-R gamma synchronization without altering the gamma power on each side (see *Figure 6—figure supplement 1*). Note the tight co-variation and L-R phase synchronization of individual gamma waves in the controls, and their outphasing after lidocaine (left tracings). Evoked fEPSPs on both sides were fully blocked after lidocaine microinjection. The autocorrelation (AC) of the Schaffer activity showed oscillatory gamma waves on both sides, with a similar power before and after lidocaine administration, whilst the cross-correlogram (CC) was drastically reduced (**D**). (**E**) The CC between L and R Schaffer activities reduces significantly (n = 4 experiments; t test). (**F**) L-R spectral coherence before and after lidocaine injection. Note the disappearance of significant bars (in blue) in the gamma band (40 Hz) after lidocaine (arrows). The effects of unilateral CA3 blockade on unit firing are in *Figure 6—figure supplement 2*.

The following source data and figure supplements are available for figure 6:

**Source data 1.** Effect of disruption of the VHC by lidocaine on the features of gamma waes.

**Figure supplement 1.** Uncoupling of CA3 gamma oscillators (additional data).

**Figure supplement 2.** Inactivation of the CA3 in one side alters individual but not population firing rates in the other.

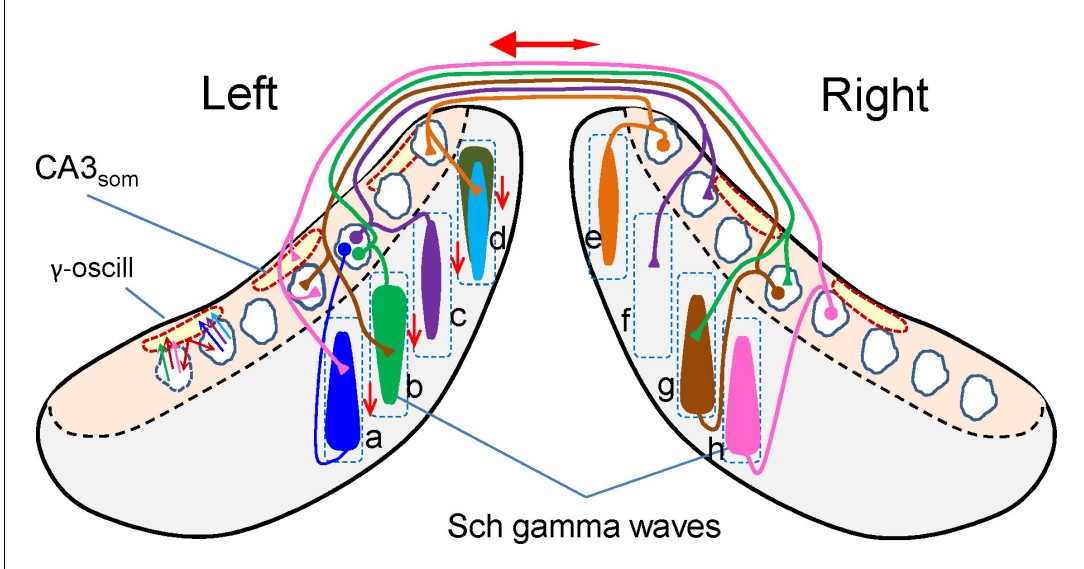

**Figure 7.** Scheme of ipsi and bilateral CA3 and CA1 relationships associated to the occurrence of Schaffer gamma waves. Cells, domains and pathways activated during an imaginary gamma cycle in a coronal representation of the two hippocampal lobes (CA1 and CA3 subfields are in grey and beige, respectively). Only anatomical and functional relationships that are relevant to the present findings are included. Dashed squares depict lamellar-like CA1 domains. Spindle-like forms represent the CA1 dendritic domains activated by a CA3 assembly, i.e.: a Schaffer gamma wave. Dashed outlines in red represent spatial domains in the CA3 soma layer inhibited by CA3$_{som}$ waves. The CA3 assemblies elicit ipsilateral Schaffer waves that are matched by near synchronous waves on the other side at roughly homotopic sites (i.e.: bilateral waves *a-h, b-g*). Excitation in the contralateral side does not produce field potentials (*Martín-Vázquez et al., 2015*), as noted by the presence of unilateral waves (e.g., *c-f*). Schaffer waves may initiate in anterior or posterior sites of the CA1 indistinctly (indicated by the wider end of the spindles), probably originated by assemblies in different CA3 zones. Very few assemblies projecting to a given CA1 domain overlap in the same cycle. For example, domain *d* receives two ipsilateral Schaffer waves (in cyan and dark green) and a contralateral (unseen) wave (in orange). As part of the intrinsic gamma oscillator, different CA3 assemblies activate the same basket cells, which inhibit many more PCs, independently of whether they belong to the firing assembly or not (amplitude mismatch between CA3$_{som}$ and Schaffer waves). The double-headed red arrow represents the overall R-to-L precedence, and the vertical arrows represent the overall anticipation of waves at anterior sites of the lamellar CA1 domains.

temporal axis by multiple independent CA3 assemblies with either a dominant lamellar or longitudinal projection, converging with others entering from the contralateral side. Such complex modular bilateral mating requires precise intra and interhippocampal control of their timings to achieve efficient global orchestration. There is insufficient data available to draw a more complete picture, and obtaining such information requires simultaneous recording over more CA1 and CA3 sites than was performed here, and additional techniques to monitor the spontaneous activity of other subpopulations not accessible from LFPs. Nevertheless, some roles of local inhibitory networks can be derived from present CA3$_{som}$ data and the available literature.

Globally, the bilateral CA3$_{som}$ relationships replicate those of Schaffer activities, although some of the differences indicate that they are less relevant for bilateral entrainment. For instance, the bilateral coherence between right and left CA3$_{som}$ activities, and the covariation of paired waves, is very small. Instead, the CA3$_{som}$ gamma wavelets are tightly locked to ipsilateral Schaffer waves. The results obtained with the phase-difference test indicate a preferred CA3$_{som}$ to Schaffer direction, possibly distorted by the frequent mismatch of waves and shorter duration of CA3$_{som}$ waves. The lag between paired waves is a more realistic estimation, and it determines that the two initiate nearly simultaneously, as seen in evoked potentials. We should have in account that the Schaffer waves are detected in CA1 recordings ~2–3 ms after CA3 PCs fire, a lag that is fully compatible with CA3$_{som}$ waves arising from basket cell-mediated recurrent inhibition following the same PC firing (*Hájos et al., 2004*). All these findings support a main participation of the CA3$_{som}$ activity in ipsilateral mechanisms, as also indicated by the large ipsilateral coherence and the nearly synchronous occurrence of waves at anterior and posterior sites. This latter feature is consistent with the reported coupling of activity in subnetworks of inhibitory cells (*Galarreta and Hestrin, 1999*; *Tamás et al.,*

*2000*), which provides modulation of CA3 output over extended domains. Since the marked gamma modulation indicates their participation in the CA3 gamma oscillator (*Fisahn et al., 1998*; *Hájos et al., 2004*; *Mann et al., 2005*; *Gulyás et al., 2010*), it would be important to determine the extent of septo-temporal coherence.

On another line, one might think that a somatic inhibitory input plays a role in selecting the CA3 domain of PC cells from which the functional assembly forming a Schaffer wave arises. However, the strikingly poor covariation between $CA3_{som}$ and Schaffer waves does not appear to be in consonance with this view. Another possibility is that multiple CA3 assemblies share a common domain, since nearby PC cells rarely fire together (*Takahashi et al., 2010*; *Matsumoto et al., 2013*). Thus they would deliver Schaffer waves of similar spatial coverage in CA1. It is also compatible with the finding that Schaffer and $CA3_{som}$ waves have a tight covariation in a longer time scale. A possible explanation of this is that both CA3 PCs and basket cells receive a common tonic input (*Glickfeld and Scanziani, 2006*; *Freund and Katona, 2007*) that codes for the duration of the gamma strings and intermissions, possibly as an energy-saving mechanism to stop the intrinsic oscillators in absence of activity and when bilateral entrainment is unnecessary.

## Functional implications of asymmetric connectivity

The extreme amplitude variability of successive Schaffer waves and the degree of bilateral covariation is of particular interest. Although the functional significance of this is unknown, some possibilities arise within a global framework that includes the oscillatory nature of the activities and their cross-checking through the commissural system.

Theoretical studies show that gamma oscillations form rapidly from small clusters of interconnected cells (*Börgers et al., 2012*). The rhythmic succession of LFP events implies a certain loss of coding performance, although the variability in amplitude exhibited by Schaffer waves destroys the repetitive character of the transmission at the cellular level as it involves a different number of firing PCs, and even a different functional assembly (*Fernandez-Ruiz et al., 2012a*, *2012b*). Indeed, coding is optimized by global oscillations within a population whose units fire sparsely (*Engel et al., 2001*; *Fries, 2009*; *Chalk et al., 2016*). From the side of target CA1 units, the gamma rhythmic CA3 input implies some sort of chunked information, as recently proposed in the cortex (*Hyafil et al., 2015*). Indeed, the optimal use of these chunks would be to meet other inputs at precise times. One of these is the contralateral homonymous input with which it maintains tight covariation in some gamma strings but not in others, albeit always in phase. This observation already indicates that in some epochs the information conveyed is similar on both sides and it differs in others, which is also supported by the bilateral out-phasing of gamma waves and the occurrence of some unilateral gamma strings upon VHC disruption.

Similarly, it is interesting that CA1 PC cells fire preferentially on bilateral waves and we ruled out the possibility that this was a mere response to larger ipsilateral inputs. Curiously, the blockade of the contralateral CA3 input modifies the firing rate of units but the changes were balanced at the population level, in agreement with previous findings (*Zemankovics et al., 2013*; *Middleton and McHugh, 2016*). Similar population balance of unitary changes has been reported following long-term synaptic plasticity (*Martin and Shapiro, 2000*; *Dragoi et al., 2003*; *Yun et al., 2007*; *Fernández-Ruiz et al., 2012b*). In the context of bilateral integration, we may then infer that the two CA3-originated excitations summed in CA1 units do not operate as a simple integrator, but rather the output is defined on a single cell basis in concert with other inputs, such as that from inhibitory neurons that are also modulated by interhippocampal disruption. The rather disperse firing of PCs over the gamma cycle, in clear contrast to the phase-locked firing of putative interneurons, supports this concept. Such dispersion may arise from the slower dynamics of intracellular EPSPs compared to EPSCs (reflected in extracellular field waves), and/or the variable influence of different interneuron subtypes with distinct phase coupling to gamma waves (*Hájos et al., 2004*; *Tukker et al., 2007*; *Vinck et al., 2010*; *Varga et al., 2014*). Indeed some interneurons can be envisaged as timing devices for different inputs to converge at optimal instants. In turn, the gamma-phase shifting of CA1-PCs is in contrast with the tight gamma coupling of CA3-PCs (*Hájos et al., 2004*; *Fernandez-Ruiz et al., 2012a*), indicating a short-lived gamma pattern restricted to this segment of the hippocampal circuit. Indeed, we have not found a gamma pattern in the subicular projection of the CA1 population (Makarova and Herreras, unpublished results).

Theoretical studies indicate that connection asymmetry favors temporal association (*Sompolinsky and Kanter, 1986*) and the generation of robust transients of sequential activation (*Rabinovich et al., 2008*). The dominance of the right side of the CA3 may thus constitute a physiological mechanism to provide a fast dynamic trade-off between left and right CA3 gamma oscillators for the fast bilateral synchronization and stabilization of the entrainment of autonomous CA3 gamma oscillators despite of the strong amplitude variability of individual waves. Overall, the present observations are compatible with the hypothesis that the main functional role for CA3 gamma oscillations is their bilateral entrainment, rather than transmitting gamma activity to the cortex.

### Asymmetric connectivity and lateralized routing of information: implications for coding and behavior

It is important to differentiate between asymmetric connectivity and lateralized routing of information. The larger Schaffer waves in the right-hand side may reflect left-right asymmetry in the subunit composition of Glutamate receptors reported in the CA1 (*Kawakami et al., 2003*; *Shinohara et al., 2008*), although a larger size of CA3 assemblies on the right-hand side cannot be excluded. In addition, right dominance better fits a hardware-like role to promote the integration of lateralized streams of information, but we cannot rule the priority or the qualitative relevance of that running through the left and right circuits during specific tasks (*Klur et al., 2009*; *Shinohara et al., 2013*; *Shipton et al., 2014*). Different hippocampal subregions are known to be differentially involved in the encoding and consolidation/retrieval of spatial information (*Jerman et al., 2006*; *Hunsaker et al., 2008*), or in perceived versus reconstructed scenes (*Zeidman et al., 2015*). Also, spatial and non-spatial information converge in the hippocampus to form episodic memories (*Leutgeb et al., 2004*), although functional difference between hemispheres has rarely been invoked. A significant finding is that both CA3 are required for short-term memory, yet only inactivation of the left CA3 impairs performance in an associative spatial long-term memory task and plasticity (*Shipton et al., 2014*). Along with the present observations of independent gamma strings upon VHC blockade, it becomes clear that the left and right sides do not convey equivalent information. Whether lateralization is devoted to different sensory modalities, to features of a scene, or to the perceived/recalled nature remains unknown.

The hippocampus is thought to represent behavioral episodes as sequences of events (*Eichenbaum, 2004*). How these events are coded at the cellular level is a matter of intense study, although experimental evidence that is largely based on unitary studies is hard to unite in a single framework. Based on the assembly code of CA3-transmitted activity, one interesting possibility is that groups of CA3 assemblies distributed over the septotemporal axis (*Benito et al., 2014*) sequentially code the relevant features of a scene in a gamma string, which are checked against each other bilaterally to form higher order features and to modulate parallel or lateralized output to the cortex (*Ciocchi et al., 2015*) (*Figure 7*). The simplicity of this mechanism makes likely that it subserves bilateral assemblage of lateralized activity in other brain structures.

## Material and methods

The experiments were performed in accordance with European Union guidelines (2010/63/CE) and Spanish regulations (BOE 67/8509–12, 1988) regarding the use of laboratory animals. The Research Committee at the Cajal Institute approved all the experimental protocols.

### Experimental procedures

The animals were anesthetized with urethane (1.2 g/kg, i.p.), fastened to a stereotaxic frame (Narishige, mod. SR-6R) and the body temperature was maintained at 37°C with a heating pad and feedback control. The long-lasting anesthetic bupivacaine (0.75%) was applied at surgical wounds. In experiments aimed at exploring the inter- and intrahippocampal Schaffer gamma (n = 7 animals) one or two concentric bipolar stimulating electrodes were placed in the soma layer of the CA3b region in the left or in both hemispheres for orthodromic activation of the CA1 (AP 2.9; L ± 2.6; V 3.4 mm from Bregma and cortical surface). Recordings were obtained from a total of 64 sites by two twin-shank linear probes (16 channels per shank, 100 µm intersite distance: Neuronexus, Ann Arbor, MI) that were located at homotopic sites of the dorsal hippocampus across the CA1 region and that also spanned the DG/CA3 (AP 4–4.5; L ± 2.6 mm) (*Figure 1B*). The shanks (0.5 mm apart) were oriented

parallel to the midline to maximize spatial coherence of Schaffer activity (*Benito et al., 2014*). In a different group of experiments aiming to disrupt the commissural fibers (n = 4 animals) two single shank probes were used (32 channel each, 50 µm intersite distance) (AP 4.5; L ± 2.6). CA1 units were obtained from the recordings in the first animal group as well as from an additional group of 3 animals using four-shank probes (150 µm apart), each with two tetrodes separated vertically by 200 µm. The probes were soaked in DiI (Molecular Probes, Invitrogen, Carlsbad, CA) before insertion for postmortem evaluation of their placement in histological sections. A silver chloride wire in the neck skin served as a reference for recordings. Signals were amplified and acquired using Multi-Channel System (Reutlingen, Germany) recording hardware and software (50 kHz sampling rate).

The commissural pathway was functionally blocked by local application of the Na-channel blocker lidocaine (HCl 2%: Braun) to the left side of the VHC (AP 1.5; L 0.5 mm). We injected a microdrop of drug solution (~50 nl) through a glass pipette (7–12 µm at the tip) using pressure through a syringe connected by a plastic tube (*Mizumori et al., 1989*) (Picospritzer, General Valve). The same pipette was also employed for electrical stimulation, and the bilaterally recorded CA1 evoked fEPSPs guided the placement of the linear probes, as well as the effectiveness of the drug. Placement stability of the VHC pipette was weak upon injection. Therefore, we only used experiments in which it remained in site after delivery of the drug and the evoked potentials were abolished for at least 2–3 min.

The design described above is not sufficient to study the effects of interhippocampal disruption on the spontaneous unit firing of CA1 cells, which requires longer epochs. To this end in another group of experiments (n = 4) we used the selective blockade of the CA3 itself on the left-hand side, sparing the circuitry on the right-hand side. Larger microdrops (0.1–0.2 µl) of lidocaine were injected into the septal pole of the left CA3 through a 200 µm wide silica cannula inserted into the cannula of the concentric stimulating electrode along with the inner wire, connected through a plastic tube to a 5 µl Hamilton syringe. Admittedly, the disruption of the interhippocampal communication through the VHC may not be complete since fibers originating in dorso-caudal CA3 sites running into the fimbria (*Laurberg, 1979*) are spared by the injection. The extension of the drug was monitored by the modulation of evoked potentials elicited by the ipsi- and/or contralateral CA3, whose fibers pass above the injection site on their way from the fimbria to the CA1. Typically 1 or 2 injections were sufficient to ensure complete blockade of ipsilateral CA1 fEPSPs that was stable for at least 10 min. For longer drug action, successive microdrops were injected at 5 min intervals, resulting in reasonable stability as witnessed by the selective steady effect on ipsilateral CA1 fEPSPs. Although evoked activity began to recover 15–30 min after injection, we noted that hippocampal LFP activity displayed abnormal interregional patterns for long periods (>60–90 min). Hence, a recovery period could not be reasonably established for unitary analysis.

At the end of the recording session the animals were sacrificed by anesthetic overdose, and their brain was removed and maintained in 4% paraformaldehyde in saline. Sagittal brain sections (100 µm) were stained with bis-benzimide and the electrode positions assessed by fluorescence microscopy (*Figure 2—figure supplement 1*).

## Data analyses

### Obtaining Schaffer-specific and CA3-somatic LFP activities through an independent component analysis

Layer-specific gamma activity has been seen in the hippocampus that reflects independent synaptic inputs to target PCs and granule cells (*Makarov et al., 2010*; *Fernandez-Ruiz et al., 2012a*). Only pathway-specific LFPs have unique time dynamics that can be interpreted as a postsynaptic convolution of the spike output in a homogeneous upstream population (*Herreras et al., 2015*). However, due to the far reach of field potentials from currents sources, the features of individual events recorded at a single site contain a mixed voltage contribution from several sources and cannot be safely assigned to a specific pathway. To obtain the precise temporal dynamics of CA3 PC output we used the Schaffer LFP activity in CA1 of each side. Other field potential contributions added by local or remote co-activated pathways were removed using the ICA (*Herreras et al., 2015*) assuming spatial constancy of the electrical fields created by the synaptic inputs. The pathway-specificity of the so obtained CA3 Schaffer input to CA1 has been already verified (*Korovaichuk et al., 2010*; *Fernandez-Ruiz et al., 2012a*). Moreover, we recently showed that the contralateral (commissural) CA3 input to CA1 produces negligible LFPs due to a combination of geometrical and functional factors

(*Martín-Vázquez et al., 2015*). Schaffer LFPs can thus be considered to reflect ipsilateral CA3 output only, facilitating a comparison of the ipsi and contralateral CA3 dynamics by obtaining the Schaffer component in both sides.

We note that although the Schaffer and CA3$_{som}$ activities display high power in the low-gamma frequency band (30–50 Hz) the individual waves may have shorter or longer duration than the oscillation wavelength. In the CA3 region itself we aimed to separate the somatic gamma waves that were earlier shown to correspond to inhibitory somatic currents (*Benito et al., 2014*), possibly from basket cells (*Fisahn et al., 1998*; *Hájos et al., 2004*; *Mann et al., 2005*; *Gulyás et al., 2010*). The CA3$_{som}$ waves were recorded with the same shanks as the Schaffer waves in CA1.

The ICA considers recorded LFP signals $u_m(t)$ as the weighted sum of the activities of $N$ neuronal sources or LFP-generators:

$$u_m(t) = \sum_{n=1}^{N} V_{mn} s_n(t), \quad m = 1, 2, ...M \quad (1)$$

where $(V_{mn})$ is the mixing matrix composed of the so-called voltage loadings or spatial weights of $N$ LFP-generators on $M$ electrodes and $s_n(t)$ is the time course of the *n-th* LFP-generator. Thus, the raw LFP observed at the *m-th* electrode tip is a linear mixture of the electrical activity of several independent LFP-generators. Using $u_m(t)$ the ICA finds both $(V_{mn})$ and $s_n(t)$. The joint curve of spatial weights of an LFP-generator $(V_n)$ reflects the instant depth profiles of the proportional voltage among the sites, while the time-course $s_n(t)$ can be considered as a postsynaptic temporal convolution of spike output in an afferent population (i.e. afferent spike trains). The mathematical validation and practical limitations of this approach, as well as the possible sources of cross-contamination have been investigated thoroughly using realistic modeling (*Makarova et al., 2011*; *Fernandez-Ruiz et al., 2013*; *Martín-Vázquez et al., 2013*; *2015*) (for a review see Ref. *Herreras et al., 2015*). In this study we employed the kernel density ICA algorithm (*Chen, 2006*), customarily implemented in MATLAB (MathWorks). Normally few LFP-generators (four to seven) exhibited significant variance and distinct spatial distributions (*Figure 1D*), which permits further optimization by pre-processing the LFPs prior to performing the ICA through dimension reduction using the principal component analysis (PCA). This approach efficiently diminishes the presence of noisy weak generators. The PCA also stabilizes and accelerates the subsequent convergence of the ICA. This analysis can be performed with our LFP-sources software, freely available at http://www.mat.ucm.es/~vmakarov/downloads.php. Given the small variance contributed by the Schaffer or CA3$_{som}$ LFPs to CA1/DG profiles (*Martín-Vázquez et al., 2015*) the optimal separation may require different strategies as to the selection of recordings channels making up the data matrix, depending on the features of the spatial gradients captured by the linear arrays (*Figure 1—figure supplement 1*). We earlier showed that the separation of blended sources that share a high percentage of co-varying time-points (e.g, a high content of phase-locked gamma waves in both) benefits from channel selection, otherwise it is not necessary (*Makarova et al., 2011*; *Martín-Vázquez et al., 2015*). Different gross electrographic patterns may therefore require one or another strategy to build-up the data matrix.

The most characteristic pattern of both Schaffer and CA3som activities observed is gamma strings of variable duration and prevalence. These are intermingled with SPWs (*Csicsvari et al., 1998*) that are known to occur synchronously in both hemispheres (*Carr et al., 2012*) and that were not further studied here. Besides, since SPWs have a strong contribution to the variance of the signal we removed them manually to avoid bias in the bilateral comparisons of Schaffer gamma strings. Slow artifacts were removed from the original LFPs through a high-pass filtering (1 Hz cut-off frequency), yet no further signal accommodation was performed.

## Estimating the synchronization between LFP-generators

Coarse synchronization between different recording sites and time courses of LFP-generators was estimated using the cross-correlation coefficient (CC) and spectral coherence. The CC was obtained as:

$$R = \frac{C_{12}}{\sqrt{C_{11} C_{22}}} \quad (2)$$

where $(C_{ij})$ is the covariance matrix of two random variables. We used epochs of 80 s for analysis. Spectral coherence was calculated by:

$$C_{xy}(f) = \frac{|P_{xy}(f)|^2}{P_{xx}(f)P_{yy}(f)} \tag{3}$$

where $(P_{ij}(f))$ is the matrix of cross-power spectral density. To determine the level of significance we used the surrogate data test (**Schreiber and Schmitz, 2000**). Randomizing phase relations and keeping other first order characteristics intact, we obtained surrogate time series from the original signals. For each experiment we generated 400 surrogates and we evaluated pairwise spectral coherences. The level of significance (at $\alpha$ = 0.05) was then calculated for each frequency value and coherence above this level was considered statistically significant.

To explore the functional connectivity between LFP generators, we used two different tests. First, we used Granger causality (GC), a statistical test that is widely used to infer directional connectivity among brain areas (**Friston, 2011**). It is based on examining the coefficients of a regression model obtained from two time series that are delayed one with respect to the other (**Granger, 1969**). To calculate GC, we employed the MVGC toolbox for Matlab (**Barnett and Seth, 2014**). As input time series, we used the activation signals of the right and left Schaffer LFP-generators.

To confirm the results obtained by the analysis of individual Schaffer waves (see below), in another test we estimated the phase shift between the right and left sides based on the calculation of the phases of Schaffer activations. We applied the Hilbert transform to left and right Schaffer activities (implemented in Matlab) and we evaluated the phase dynamics of both generators, $\phi_L(t)$ and $\phi_R(t)$. We then found time instants $t_i$ of zero crossing from above of the phase of the R generator (i.e., $\phi_R(t_i)=0$ and $\phi'_R(t_i)<0$). Such time instants correspond to the beginnings of negative pulses (fEPSPs) in the R Schaffer. Finally, for these time instants we found the phases in the left Schaffer generator, $\{\phi_L(t_i)\}$. If there is indeed a time lag between the right and left sides, then the distribution of these phases plotted in a probability distribution histogram (PDH) should have a statistically significant peak at positive phase values. Since phases are defined in a circle, to test the statistical significance of the peak and to evaluate the mean value of the phase difference we employed the circular statistics toolbox for Matlab (**Berens, 2009**). Then the time lag was calculated by evaluating the ratio of the phase lag over the mean circular frequency of the L generator.

## Retrieval and quantification of gamma waves

In order to compare the features of individual gamma waves and evaluate their synchronization at different sites, accurate determination of the waveforms is required (start time, amplitude and duration). Earlier we have used a wavelet transform method to detect short pulses (gamma waves) in the signal (**Martín-Vázquez et al., 2015**). However, we noticed that due to the frequent overlap of successive waves, the duration and the amplitude could be underestimated. Moreover, $w_k$ the starting time, which is essential in this study to explore bilateral synchronization, was only roughly approached. Here we implemented a new method using deconvolution of LFPs (**Figure 8**).

The method employs the modeling of the time course of a LFP generator, $s(t)$, as a weighted sum of $K$ single LFP events (so-called waves) of the pulse-like form:

$$s(t) = \sum_{k=1}^{K} w_k f(t - \tau_k; \delta_k), \qquad f(t;\delta) = H(t)\frac{t}{\delta^2}e^{-t^2/2\delta^2} \tag{4}$$

where $w_k$, $\tau_k$, $\delta_k$ are the relative weight, starting time, and time scale of the $k$-th event, respectively, and $H(t)$ is the Heaviside step function. We aim at estimating the parameter set $(w,\tau,\delta)$ from the observation of $s(t)$. To accomplish this task we use the method of maximization of the loglikelihood provided by

$$L = \sum_{n=1}^{N} \ln\left[\sum_{k=1}^{K} w_k f(t_n - \tau_k; \delta_k)\right] + \left(1 - \sum_{k=1}^{K} w_k\right) \tag{5}$$

where $t_n$ are time samples. Since the sum over components appears inside the logarithm, there is no closed form solution for maximum likelihood. Thus, to maximize $L$ we use the expectation-maximization algorithm adapted for the given likelihood function. To start the numerical scheme and to facilitate the convergence at the beginning we perform an initial estimation of the parameters by wavelet

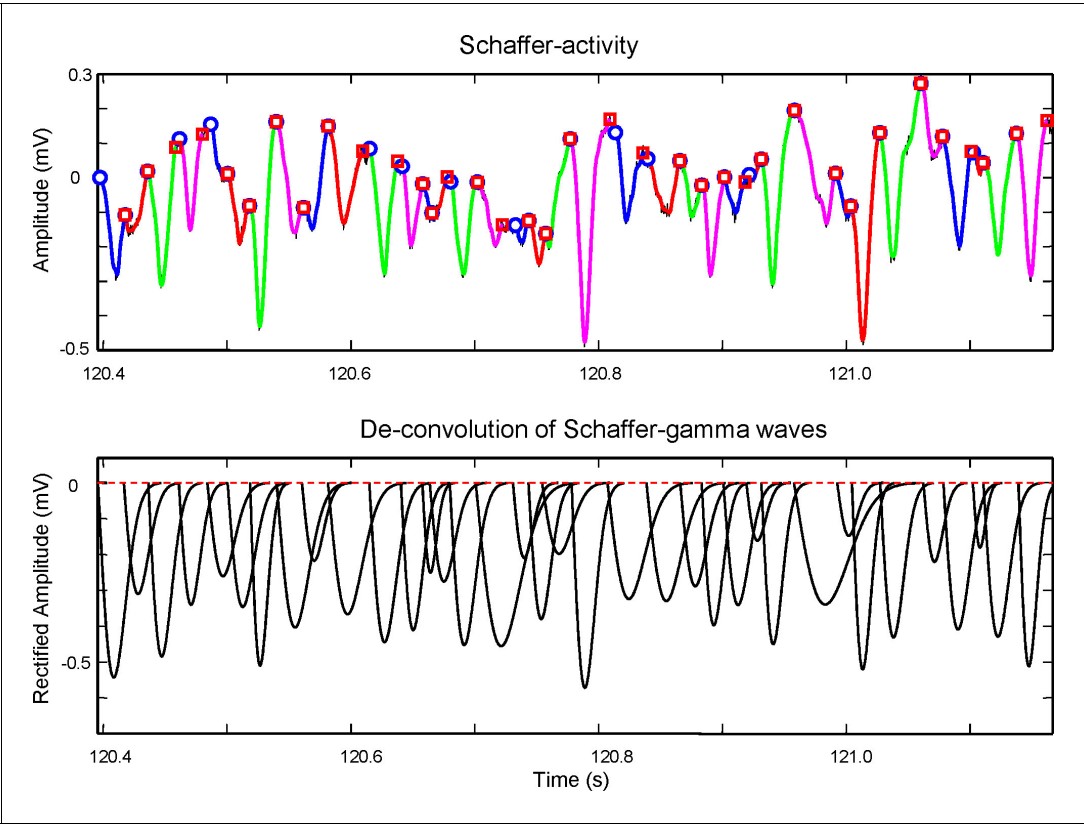

**Figure 8.** Method for the extraction of wave features from the time course of the Schaffer generator. Top panel: An epoch of the time course of the Schaffer LFP-generator (black curve) with superimposed gamma waves detected by means of wavelet decomposition (colored curves). Each colored curve has its start and end points marked by a circle and square, respectively. The curves are used to evaluate the initial conditions for the maximization of likelihood method. Due to uncertainty in the baseline the initial weights (amplitudes) of gamma waves are taken as equal. The start (ts, circle) and end time (te, square) of a curve provide initial values for tk = ts and δk = te – ts. The optimization process then corrects these values by maximizing the loglikelihood. Bottom: Deconvoluted gamma waves found by the method. By adding them together we obtain the time course of the Schaffer generator shown in the top panel. The amplitude, A, is measured as the peak value from a baseline obtained by weighted interpolation of the start times. The method to associate gamma waves and spike firing is shown in *Figure 8—figure supplement 1*.

The following figure supplement is available for figure 8:

**Figure supplement 1.** Analysis and statistics of left and right gamma waves and their relation to spike production in CA1 units.

decomposition (*Hramov et al., 2014*). *Figure 8* (top panel) shows an example of the identification of LFP-events in an epoch of activity of the Schaffer generator. This enables the number of gamma waves to be estimated, as well as their starting times and durations. Initial weights are taken the same $w_k = 1/k$ and then an iterative process starts and successively maximizes the likelihood. Once a maximum has been found, we get all necessary parameters to describe the gamma waves. The events that make up the time course of the Schaffer generator are shown in *Figure 8* (bottom panel).

In each experiment we identified and described the main features of gamma waves in two spatial locations of each CA1. To avoid noisy events being considered as gamma waves we set a low threshold for detection at 5 ms and 20 μV that is about 10% the mean amplitude of gamma waves across experiments. We then classified gamma waves as paired and unpaired and the criterion on pairing waves was based on the degree of temporal overlapping:

$$\Omega = 2\frac{\Delta_{ovlp}}{d_1 + d_2} \tag{6}$$

where $\Delta_{ovlp}$ is the overlapping time interval of two nearest in time waves and $d_1, d_2$ are the time durations. If $\Omega > 0.7$, then two waves are considered paired, otherwise the waves are unpaired.

We next cross-correlated the separated gamma waves obtained in two locations. The amplitudes of paired waves in different shanks were further plotted and the CC were evaluated as in (3) (e.g., for amplitudes in La and Lp sides in *Figure 2B*). To obtain the best linear fit we used the orthogonal linear regression based on PCA, since both variables had measurement error. The same procedure was applied for quantification of other characteristics of paired waves (e.g., the time lag of the bilateral pairs relative to their difference in amplitude: *Figure 2E*).

To test the population statistics on the CC (*Figure 2—figure supplement 2*) we used the statistic:

$$T = \frac{(r_{13} - r_{23})\sqrt{(n-1)(1 + r_{12})}}{\sqrt{2K\frac{n-1}{n-3} + \frac{(r_{23} + r_{13})^2}{4}(1 - r_{12})^3}} = t(n-3), \quad K = 1 - r_{12}^2 - r_{13}^2 - r_{23}^2 + 2r_{12}r_{13}r_{23} \tag{7}$$

where $r_{12}$ is the CC between Ra and Lp, $r_{13}$ is the CC between Ra and La, and $r_{23}$ is the CC between La and Lp. This allows the null hypothesis that the bilateral CC between Ra and La is equal to the unilateral CC between La and Lp to be tested.

To evaluate the confidence interval for the mean time lag between bilateral waves (*Figure 2—figure supplement 2*) we used t-statistics:

$$T_{lag} \in x \pm t_{n-1;1-\alpha/2} \frac{s}{\sqrt{n-1}} \tag{8}$$

where *x* and *s* are the mean and uncorrected standard deviation of the time lag, respectively.

## Considerations on the baseline

From the theoretical grounds pertaining to the ICA, we implicitly assume that any fluctuations contained in a given LFP-generator are debugged of volume-conducted contributions by other sources (*Herreras et al., 2015*). Hence, for the Schaffer activity, the time course *s(t)* is produced by spatiotemporal summation of excitatory currents elicited in CA1 PCs by CA3 inputs alone, while the CA3$_{som}$ generator results from summation of inhibitory currents at CA3 somata and proximal dendrites. In the latter the mean duration of waves was significantly shorter than the cycle period, thus the baseline was readily recovered and the start time of individual waves easily determined. However, the longer duration of Schaffer waves promotes unknown overlap amongst successive waves in a gamma string that hampers accurate determination of start and end times. Even slow envelopes can also be seen that may have an alternative origin in the asynchronous input from origin cells firing out of the gamma-organized assemblies. We noticed two possible sources of error. On the one hand, the method uses an interpolated baseline for overlapped waves that bias amplitude and duration towards smaller waves (these are estimated larger and longer than they in fact are: note the apparent amplitude equalization between the raw and model waves). On the other hand, the gamma waves were fitted to α-function waveforms that disproportionately overestimate the tail. To lessen the impact of these possible errors, we adjusted the duration of the numerically fitted waves using a reduction factor obtained in each experiment from 50 waves chosen randomly (0.7–0.8 for Schaffer and 0.4–0.5 for CA3$_{som}$). Yet, since the bulk of results shown in this study are based on the start times of paired waves that are barely affected by the chosen baseline, the main conclusions hold whatever buildup for the baseline is considered.

## Spike sorting and classification

Spike trains of individual units were obtained from unfiltered recordings using two methods: the wavelet-enhanced PCA spike sorting (*Pavlov et al., 2007*) and local CSD method. Units were classified into two subclasses, pyramidal cells and putative interneurons, according to the location of the recording site (within or outside the pyramidal body layer) and based on additional standard electrophysiological criteria (*Csicsvari et al., 1998*): (i) spike width (>0.4 ms for pyramids and <0.4 ms for putative interneurons); (ii) mean firing rate (<5Hz vs. >5Hz); (iii) relation to theta rhythm (the firing rate decreases for pyramidal cells and increases or remains unchanged for interneurons); (iv) firing pattern (isolated spikes vs. bursting); (v) the presence of complex spikes (in pyramidal cells only); and (vi) the decay of autocorrelograms (fast vs. slow).

## Relation of spike firing to excitatory gamma waves

The relation of gamma waves to cell firing was studied by two methods. First, we built comodulograms in which the lag from the beginning of the wave on both sides was plotted against the spike time, and it was represented in time-to-time-density bi-dimensional histograms.

In *Figure 5—figure supplement 1C* (upper subplots), the preferred intervals for combined L and R input leading to spike firing appear as X-Y surfaces coded from blue (no occurrences) to red (a high density of occurrences). Values lying in or near the diagonal indicate spike firing upon coincident R and L inputs, while accumulation of values parallel to one axis indicate the preferred driving of spikes according to the input in the corresponding side. Parallel diagonal bands appeared in some cells that denote chance firing subsidiary to dominant gamma repetition of LFP waves. In order to establish the level of statistical significance for the observed spike density (upper plots) we introduced a probabilistic model for chance coincidence and then subtracted the chance value from the observed density and only plotted the positive values (lower plots):

$$H_{sgn} = H - H_{0.95} \tag{9}$$

To evaluate the significance level $H_{0.95}$ we assumed that LFP events in the left and right sides were generated by Poisson random processes and the joint probability to having at least one L and one R event within time window $\Delta$ can be represented as the product of marginal probabilities:

$$P = \left(1 - e^{-N_l\Delta/T}\right)\left(1 - e^{-N_r\Delta/T}\right) \tag{10}$$

where $N_l$ and $N_r$ are the number of events that occurred over the observation time $T$. Assuming that the number of recorded spikes $N_s >> 1$ but the firing rate is sufficiently low, we can write the statistical distribution:

$$Y = Bin(N_s, P) \approx N(PN_s, \sqrt{N_s P(1-P)}) \tag{11}$$

Thus, the threshold for the 2D histogram is given by:

$$H_{1-\alpha} = PN_s + \lambda_{1-\alpha}\sqrt{N_s P(1-P)} \tag{12}$$

A second test aimed to quantify how well spike firing correlates with the occurrence of unilateral or bilateral waves, or whether it was independent of gamma firing (*Figure 8—figure supplement 1*). For each recording, the time line was divided into four types of time-windows according to the presence of gamma waves induced by the left or right Schaffer generators: (1) None, i.e.: no gamma waves on any side; (2) R, i.e.: a wave is present on the right side only; (3) L, i.e.: wave is present on the left side only; and (4) Bilateral, i.e.: gamma waves on both sides. Then each spike in a spike train falls into one of these classes of time-windows. For each spike train we calculated the number of spikes of different classes $\{q_1, q_2, q_3, q_4\}$ and we evaluated the ratios of the total length of the four classes of time-windows to the recording time $\{r_1, r_2, r_3, r_4\}$. For statistical inference, as a model of a random spike train (i.e.: not dependent on field potentials) we used a train generated from an independent Poisson distribution. For such a train the number of spikes falling inside time-windows of the $i$-th type follows binomial statistics $Y_i \sim Bin(n, r_i)$. Therefore, we can establish the confidence limits for the number of spikes of a random spike train $\left(nr_i \pm \lambda_{0.95}\sqrt{nr_i(1-r_i)}\right)$, where $\lambda_{0.95}$ is given by the normal distribution. If $q_i$ exceeds these limits, then we can conclude that the number of spikes of the given class in the train cannot be explained by mere coincidence.

## Distribution of PC firing over the power of the Schaffer activity

First we note that an LFP-generator (e.g., Schaffer) has a certain power distribution $f(p)$, where $p(t) = s^2(t)$ is the instantaneous power. Then, any random spike train will have the same distribution of spikes over the power of the LFP-generator, since it will represent a random sampling from $p(t)$. Therefore, for a given spike train $\{t_i\}$ of a pyramidal cell, we evaluated its power distribution from the sample $\{p(t_i)\}$ and then subtracted the baseline distribution $f(p)$ corresponding to a random train. The differential distribution obtained describes the deviation of the given spike train from a random one. Positive values indicate a higher firing rate at the corresponding power of the LFP-generator, while a negative differential distribution points to less frequent PC firing relative to a random train.

For each PC train we divided all the spikes into four groups: (1) spikes coinciding with unilateral gamma waves on the right side; (2) spikes coinciding with unilateral gamma waves on the left side; (3) spikes coinciding with bilateral gamma waves; and (4) the remaining ones. For spikes of the first three classes we evaluated the differential power distributions. Subsequently, we evaluated the mean differential distributions over all available PCs and estimated the confidence intervals.

## Acknowledgements

We thank D van Helden, P Varona, G Perea, S Canals, and M Morales for helpful comments and Mark Sefton (BiomedRed) for editorial support. This work was supported by the grants BFU2013-41533R (OH) and FIS2014-57090-P (VAM) from the Spanish Ministry of Economy and Competitiveness, and 15-12-10018 from the Russian Science Foundation (VAM, development of numerical methods). NB and GM-V held PhD scholarships FPI-BES-2008-009563 from the Spanish Ministry of Science and Innovation and FPU-AP2010-1278 from the Spanish Ministry of Education. The authors declare no competing financial interests.

## Additional information

### Funding

| Funder | Grant reference number | Author |
| --- | --- | --- |
| Ministerio de Economía y Competitividad | BFU2013-41533R | Nuria Benito<br>Gonzalo Martín-Vázquez<br>Julia Makarova<br>Oscar Herreras |
| Spanish Ministry of Science and Innovation | FPI-BES-10752008-009563 | Nuria Benito<br>Gonzalo Martín-Vázquez |
| Spanish Ministry of Science and Innovation | FPU-AP2010-1278 | Nuria Benito<br>Gonzalo Martín-Vázquez |
| Russian Science Foundation | 15-12-10018 | Valeri A Makarov |
| Ministerio de Economía y Competitividad | FIS2014-57090-P | Valeri A Makarov |

The funders had no role in study design, data collection and interpretation, or the decision to submit the work for publication.

### Author contributions

NB, GM-V, Acquisition of data, Analysis and interpretation of data, Drafting or revising the article; JM, Analysis and interpretation of data, Drafting or revising the article; VAM, Acquisition of data, Analysis and interpretation of data, Drafting or revising the article, Contributed unpublished essential data or reagents; OH, Conception and design, Analysis and interpretation of data, Drafting or revising the article

### Author ORCIDs

Oscar Herreras, http://orcid.org/0000-0002-8210-3710

### Ethics

Animal experimentation: The experiments were performed in accordance with European Union guidelines (2010/63/CE) and Spanish regulations (BOE 67/8509-12, 1988) regarding the use of laboratory animals. All of the animals were handled according to approved institutional Bioethics and Biosecurity Committee of the CSIC (ref:15/10/2014). The Ethics Committee for Animal Research at the Cajal Institute approved all the experimental protocols (Ref. CEEA-IC2011/011/CEI3/20131213). All surgery was performed under urethane anesthesia, and every effort was made to minimize suffering.

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
