## [Decision Letter]

Thank you for submitting your article "The right hippocampus leads the bilateral integration of γ-parsed lateralized information" for consideration by *eLife*. Your article has been favorably evaluated by Timothy Behrens (Senior Editor) and three reviewers, one of whom, Marlene Bartos (Reviewer #1), is a member of our Board of Reviewing Editors.

The reviewers have discussed the reviews with one another and the Reviewing Editor has drafted this decision to help you prepare a revised submission.

Summary:

By using independent component analysis (ICA) and wavelet-based deconvolution of γ activities in the hippocampal areas of rats the authors identified a significant dominance in the amplitude of γ oscillations in the right hippocampal CA1 and CA3 areas compared to the left regions. These data fit to previous investigations showing lateralized gene expression, spine size, and synaptic plasticity. Γ activity is very synchronous between the two CA3 areas due to their tight mutual coupling whereas a γ phase shift was observed in CA1. Principal cell activity in CA1 is associated with bilateral γ waves whereas interneuron activity was associated to unilateral γ patterns. These findings were considered by the reviewing editor and the two reviewers as potentially very important based on detailed state-of-the-art analysis of local field events. However, some concerns emerged during the discussion among the reviewing editor and the two reviewers which need to be considered by the authors during the revision phase:

Essential revisions:

1) Please confirm that the apparent phase difference between both hippocampi is real and not due to differences in γ waveforms on the two sides by using an additional way of extracting phase difference such as by Granger causality or transfer entropy of the ICA-derived signals.

2) Since the entire study depends on the validity of the ICA method, all reviewers ask the authors for additional simultaneous dual recording from both CA3 regions to extract the proposed contralateral γ propagation.

3) The lag time between right and left CA1 LFPs is rather small (0.67 ms). Please rule out the possibility that the difference might be due to a small anatomical asymmetry between the two sides. Please measure and report the electrode implantation sites relative to anatomical landmarks on the two sides to confirm the recording sites are indeed homotopic.

4) Pyramidal cell firing is associated with bilateral γ, which it is interpreted as bilateral summation whereas interneuron firing is associated with unilateral γ. Since bilateral γ is associated with higher power, it is unclear whether the association with bilateral γ is simply a consequence of increased power, or of coincidence of γ waves. To distinguish between these possibilities, the authors should correlate firing with power under unilateral and bilateral conditions.

5) Report effects of disruption of the ventral hippocampal commissure on principal cell and interneuron activity.

[Editors' note: further revisions were requested prior to acceptance, as described below.]

Thank you for resubmitting your work entitled "The right hippocampus leads the bilateral integration of γ-parsed lateralized information" for further consideration at *eLife*. Your revised article has been favorably evaluated by Timothy Behrens (Senior editor), a Reviewing editor, and one reviewer.

The manuscript has been improved but there are some remaining issues that need to be addressed before acceptance, as outlined below:

After careful re-assessment of the manuscript, the reviewing editor and one reviewer agreed that the authors performed a very good revision. However, we would like to ask you to be more precise in the usage of terminologies. In particular, certain essential terms like 'γ strings' and 'γ waves' have not been quantitatively defined. It would be an advantage if the authors could define each term quantitatively (e.g. what frequency and amplitude criteria were used to identify γ waves, and how long would a sequence of γ waves need to be in order to be termed a string?). It is unclear why the authors choose one term over another. For example, what is the difference between a 'γ event' and a 'γ wave', or between a 'string' and a 'thread'?

---

## [Author Response]

Essential revisions:

*1) Please confirm that the apparent phase difference between both hippocampi is real and not due to differences in γ waveforms on the two sides by using an additional way of extracting phase difference such as by Granger causality or transfer entropy of the ICA-derived signals.*

Two additional tests have been incorporated, Granger Causality and a phase-difference test. Both corroborate the overall right-hand leading obtained with extracted paired waves (subsection “Schaffer gamma activity is stronger and precedes in the right side”, last two paragraphs; subsection “Tight ipsilateral synchrony but weak bilateral entrainment characterizes the CA3 gamma waves in the soma layer”, fourth paragraph and Figure 3).

*2) Since the entire study depends on the validity of the ICA method, all reviewers ask the authors for additional simultaneous dual recording from both CA3 regions to extract the proposed contralateral γ propagation.*

The same experiments contained CA3 recordings. As suggested we added this new data and performed the analysis of somatic CA3 waves in all sites and also related them to simultaneously recorded Schaffer waves (l218-276, Figure 4 and Figure 4—figure supplement 1). Somatic CA3 waves have strong ipsilateral relationships and followed similar bilateral lags as Schaffer waves, but exhibit very weak bilateral entrainment. We thus conclude that they are mostly serving ipsilateral mechanisms.

*3) The lag time between right and left CA1 LFPs is rather small (0.67 ms). Please rule out the possibility that the difference might be due to a small anatomical asymmetry between the two sides. Please measure and report the electrode implantation sites relative to anatomical landmarks on the two sides to confirm the recording sites are indeed homotopic.*

We reconstructed all anatomical landmarks (Figure 2—figure supplement 1). Overall, it turns out that the lags are robust for slight asymmetric positioning. Some variability between experiments (Figure 2) may indeed be due to shifted antero-posterior coordinates of the twin shanks or a slightly asymmetric location respect to the midline caused by a deficient bregma reference. Evoked potential profiles are highly precise and matched histological data. Also, additional comparisons have been completed between pairs of sites not presented in the short version that corroborate with partial lags the overall findings (subsection “Variable ipsilateral dynamics subjects to global asymmetric bilateral coupling”).

*4) Pyramidal cell firing is associated with bilateral γ, which it is interpreted as bilateral summation whereas interneuron firing is associated with unilateral γ. Since bilateral γ is associated with higher power, it is unclear whether the association with bilateral γ is simply a consequence of increased power, or of coincidence of γ waves. To distinguish between these possibilities, the authors should correlate firing with power under unilateral and bilateral conditions.*

To address this question we included an analysis exploring the relation of spike firing and the power of concomitant bilateral potentials. The results indeed confirmed that when spikes matched bilateral waves, the power was irrelevant (subsection “Bilateral Schaffer gamma waves promote firing in pyramidal cells while putative interneurons prefer a unilateral input”, last paragraph and Figure 5—figure supplement 2).

5) Report effects of disruption of the ventral hippocampal commissure on principal cell and interneuron activity.

Since the effect of lidocaine in the VHC does not last long enough for unitary analysis we performed an additional series of experiments by applying lidocaine in the left CA3 itself, which leaves intact the circuitry in the right side. We confirmed that Schaffer γ remains unchanged there, as for VHC injection. The recorded units in the right side either reduced or increased the firing rate, but globally, the population did not undergo significant changes (subsection “Disruption of the interhippocampal connection reveals lateralized gamma strings”, last paragraph and Figure 6—figure supplement 2, and discussed in subsection “Functional implications of asymmetric connectivity”, third paragraph).

We made the following additional changes:

We added data from an additional experiment on the relationship between Sch and CA3 waves in different sites and sides.

We decided to jump to the full-paper format since the new demanded data required considerable extension of the Results section; also the old supplementary file containing lengthy mathematical methods has been pasted since some methodological figures were asked to pass to the main text.

The Discussion now gathers all explanations and comments formerly dispersed over the short version, plus additional discussion demanded by reviewers, into a single standard section. We added an explanatory figure (Figure 7) summarizing the main results.

The list of references has been updated.

Figure 1 has been modified to include CA3 recordings.

The file containing the data source with extracted waves has grown around ten times. Please, advise as to the suitability for uploading.

[Editors' note: further revisions were requested prior to acceptance, as described below.]

*The manuscript has been improved but there are some remaining issues that need to be addressed before acceptance, as outlined below:*

*After careful re-assessment of the manuscript, the reviewing editor and one reviewer agreed that the authors performed a very good revision. However, we would like to ask you to be more precise in the usage of terminologies. In particular, certain essential terms like 'γ strings' and 'γ waves' have not been quantitatively defined. It would be an advantage if the authors could define each term quantitatively (e.g. what frequency and amplitude criteria were used to identify γ waves, and how long would a sequence of γ waves need to be in order to be termed a string?). It is unclear why the authors choose one term over another. For example, what is the difference between a 'γ event' and a 'γ wave', or between a 'string' and a 'thread'?*

Changes to clarify terminology:

Results, first paragraph: We now defined γ waves, and specified the frequency band in which Schaffer and CA3som activities show the characteristic oscillation according to current nomenclature (35-40 Hz: low-γ). The duration and amplitude thresholds for their inclusion in statistics are now indicated in (subsection “Schaffer gamma activity is stronger and precedes in the right side”, second paragraph and subsection “Retrieval and quantification of gamma waves”, second paragraph: 5 ms and 20 μV).

Γ event/wave. Although we agree with Paul Nunez who favors the term “event” over “wave” that has specific connotations in Physics not applicable to EEGs/LFPs, we are also aware of the widespread usage of the later term in the literature. Thus, for the shake of simplicity, we removed the term “event” at most occurrences, and in a few cases “event” was changed to “LFP event” when the unspecified duration does not warrant their belonging to a frequency band (e.g., describing math protocols).

Strings. We now stated the heterogeneous duration and unpredictable occurrence of sequences of γ waves at the beginning of Results (subsection “Schaffer gamma activity is stronger and precedes in the right side”, first paragraph) that hamper quantitative characterization in the anesthetized animal. It surely will be possibly in relation to specific tasks in active animals. Thus the term “string” is not used as a quantitative entity but generically to describe any sequence of γ waves repeating regularly at low-γ frequency.

We removed the term “thread” that we’ve used earlier in a figurative sense to refer to the pathway-specific components mixed in raw LFPs.